# Comparison of Methods to Estimate Aerosol Effective Radiative Forcings in CMIP Models

Mark D. Zelinka[1], Christopher J. Smith[2,3], Yi Qin[4], and Karl E. Taylor[1]

[1]Lawrence Livermore National Laboratory, 7000 East Avenue, L-103, Livermore, CA 94550, USA
[2]School of Earth and Environment, University of Leeds, LS2 9JT, UK
[3]International Institute for Applied Systems Analysis, Laxenburg 2361, Austria
[4]Pacific Northwest National Laboratory, PO Box 999, Richland, WA 99352, USA

**Correspondence:** Mark D. Zelinka (zelinka1@llnl.gov)

**Abstract.** Uncertainty in the effective radiative forcing (ERF) of climate primarily arises from the unknown contribution of aerosols, which impact radiative fluxes directly and through modifying cloud properties. Climate model simulations with fixed sea surface temperatures but perturbed atmospheric aerosol loadings allow for an estimate of how strongly the planet's radiative energy budget has been perturbed by the increase in aerosols since pre-industrial times. The approximate partial
radiative perturbation (APRP) technique further decomposes the contributions to the direct forcing due to aerosol scattering and absorption, and to the indirect forcing due to aerosol-induced changes in cloud scattering, amount, and absorption, as well as the effects of aerosols on surface albedo. Here we evaluate previously published APRP-derived estimates of aerosol effective radiative forcings from these simulations conducted in the 6th phase of the Coupled Model Intercomparison Project (CMIP6) and find that they are biased as a result of two large coding errors that – in most cases – fortuitously compensate.
The most notable exception is the direct radiative forcing from absorbing aerosols, which is more than 40% larger averaged across CMIP6 models in the present study. Correcting these biases eliminates the residuals and leads to better agreement with benchmark estimates derived from double-calls to the radiation code. The APRP method – when properly implemented – remains a highly accurate and efficient technique for diagnosing aerosol ERF in cases where double radiation calls are not available, and in all cases it provides quantification of the individual contributors to the ERF that are highly useful but not
otherwise available.

## 1   Introduction

The primary source of uncertainty in effective radiative forcing of the climate comes from aerosols, both through their direct impact on radiation and via modifying cloud properties. This uncertainty limits our ability to know how strongly the Earth has been forced over recent decades, which hampers our ability to confidently narrow bounds on climate sensitivity based on
the observed temperature record (Sherwood et al., 2020). It also degrades our confidence in predictions of near-term climate evolution, particularly whether and how soon dangerous global mean temperature thresholds will be crossed (Watson-Parris and Smith, 2022; Dvorak et al., 2022), the committed warming level if anthropogenic emissions rapidly decrease (Armour and Roe, 2011), and how the likelihood of extreme events occurring in many regions may change (Persad et al., 2022).

Despite its importance, aerosol radiative forcing (and forcing in general) has historically been poorly diagnosed in global climate models, though recent efforts have improved this state of affairs. Standard atmosphere-only model experiments to diagnose aerosol radiative forcing have been designed and made part of the CMIP5 and CMIP6 protocols, allowing for a relatively clean method for diagnosing aerosol ERF across models. Diagnostic approaches of various levels of sophistication have also been developed and applied to climate model output to provide consistent estimates of aerosol ERF across models.

A common method for computing aerosol ERF involves additional calls to the radiation code neglecting aerosols in the atmospheric column (Ghan, 2013), as described further below. Gryspeerdt et al. (2020) extended this method to additionally separate the indirect effect into aerosol effects on cloud droplet number concentration (the Twomey effect) and aerosol-induced adjustments of cloud fraction and liquid water path. However, aerosol-free radiation diagnostics are only available in a subset of CMIP6 models that took part in the Radiative Forcing Model Intercomparison Project (RFMIP) (Pincus et al., 2016). Moreover, these diagnostics do not separately quantify the absorption and scattering components of the direct effect or the cloud absorption, scattering, and amount components of the indirect effect.

The approximate partial radiative perturbation (APRP) technique (Taylor et al., 2007) offers another method of computing aerosol ERF. Unlike the Ghan (2013) method this does not require additional aerosol-free radiation calls but rather operates on standard monthly resolution model output available across all models. Another advantage is that it allows for a breakdown of the aerosol direct effect into absorption and scattering components and of the indirect effect into absorption, amount, and scattering components. The main disadvantages are that it is an approximate technique that may be biased with respect to more accurate methods, and that it only applies to shortwave (SW) radiation. Fortunately, aerosol direct and indirect effects primarily operate in the SW with much smaller effects in the longwave (LW), except for models with strong aerosol effects on high clouds (Zelinka et al., 2014; Smith et al., 2020). Hence, APRP allows for an efficient way to quantify aerosol forcing and its individual components, making it highly valuable for systematically inter-comparing the full suite of models performing aerosol perturbation experiments.

The APRP technique has been used to quantify aerosol ERF in models, revealing the diverse strengths of the various terms comprising it across models taking part in CMIP5 (Zelinka et al., 2014) and CMIP6 (Smith et al., 2020). These studies, however, independently implemented the technique as computer code, and made different choices that have quantitative impacts on the results (as described below). In this study, we demonstrate that the implementation of the APRP method in Smith et al. (2020) was erroneous, leading to slightly biased values of aerosol ERF compared to the correct formulation implemented in Zelinka et al. (2014). We explain and quantify the two largely compensating errors that cause the bias. We also compare the two APRP formulations to the double radiation call method to evaluate how well APRP-derived results agree with this independent technique. Finally, we provide values of aerosol ERF components that are corrected from those reported in Smith et al. (2020) and supplemented with additional models that have become available since the publication of that paper.

## 2 Data and Methods

### 2.1 Climate model simulations used

Our analysis makes use of pairs of idealized atmosphere-only climate model simulations in which sea-surface temperatures (SSTs) and sea ice concentrations (SICs) are fixed at model-specific preindustrial climatological values. Aerosol burdens are set to their preindustrial levels in the control experiment and to their present-day levels in the perturbation experiment. In CMIP6, these experiments are known as 'piClim-control' and 'piClim-aer', respectively, and present-day is interpreted as year 2014 (Pincus et al., 2016). In CMIP5, they are known as 'sstClim' and 'sstClimAerosol', repectively, with present-day interpreted as year 2000. These experiments are nominally 30 years long.

We make use of all available ensemble members of all models that performed these simulations (listed in Tables 1 and 2), with the exception of the EC-Earth3 model. This was excluded because of spurious up- and down-welling clear-sky shortwave radiative fluxes at the surface ('rsuscs' and 'rsdscs') that are determined to be erroneous and bias the APRP calculations. In particular, we identified numerous examples in which these fluxes exhibited very large values at individual grid cells and months but were surrounded in time and space by near-zero values (e.g., in winter locations with negligible insolation). The monthly fields used in our analyses are listed in Table A1, and abbreviations and other nomenclature used throughout the paper are listed in Table A2.

### 2.2 Aerosol Effective Radiative Forcing Calculations

#### 2.2.1 IPCC AR6 Definitions

One can express the total change in net (downwelling minus upwelling) radiation ($\Delta R$) between a control experiment and an aerosol-perturbed experiment (both with fixed SSTs and SICs) as the sum of effective radiative forcings due to aerosol-radiation interactions ($ERF_{ari}$) and aerosol-cloud interactions ($ERF_{aci}$), as well as changes in TOA radiation due to changes in surface albedo ($\Delta R_{alb}$) and surface temperature ($\Delta R_{T_0}$):

$$\Delta R = ERF_{ari} + ERF_{aci} + \Delta R_{alb} + \Delta R_{T_0}. \tag{1}$$

As defined in IPCC AR6 (Forster et al., 2021), $ERF_{ari}$ comprises the instantaneous radiative forcing ($IRF_{ari}$), non-cloud atmospheric adjustments that are uncoupled to any change in global surface temperature, and adjustments of clouds due to changes in the thermal structure of the atmosphere caused by absorbing aerosols (also known as the "semidirect effect"):

$$ERF_{ari} = IRF_{ari} + K^T \Delta T + K^q \Delta q + K^C \Delta C_{semidirect}, \tag{2}$$

where $K^\chi$ terms quantify the sensitivity of TOA net radiation to infinitesimal perturbations in variable $\chi$:

$$K^\chi = \frac{\partial R}{\partial \chi}, \tag{3}$$

and here and elsewhere $\chi$ may represent temperature ($T$), humidity ($q$), surface albedo ($\alpha$), or clouds ($C$). It is intended that $\Delta T$ and $\Delta q$ include only changes in tropospheric temperature and water vapor that occur independently of changes in

surface temperature, not those that occur in response to any land surface temperature change in these fixed-SST simulations. In practice it is difficult to determine which portion of the change in any field is an adjustment to aerosols versus a response that is mediated by land surface temperature changes or by circulation changes induced by said temperature changes. Hence for simplicity we follow a conservative approach, also taken by AR6, and exclude the radiative impact of land temperature changes from $ERF_{ari}$. This approach leads to positive longwave $ERF_{ari}$ values that are smaller than if this correction is not applied, making the overall net negative ERF about 5% stronger on average across models, consistent with the 5% inflation that IPCC AR6 applied to the ERF values derived in Smith et al. (2020) and Zelinka et al. (2014). Finally, the radiative impact of surface albedo changes – which we also keep separate from the ERF terms in Eq 1 – arises due to deposition of absorbing aerosol on snow and ice, changes in snow and ice cover arising from the aerosol-induced change in climate, and aerosol-induced changes in the relative amount of direct versus diffuse radiation reaching the surface.

As defined in IPCC AR6, $ERF_{aci}$ comprises the instantaneous radiative forcing due to changes in cloud liquid and ice particle number concentrations and sizes ($IRF_{aci}$), and subsequent adjustments of cloud water and coverage ($K^C \Delta C_{adjustments}$):

$$ERF_{aci} = IRF_{aci} + K^C \Delta C_{adjustments}. \tag{4}$$

Although the above decomposition of direct and indirect forcings is employed by IPCC AR6 (Forster et al., 2021), in the Ghan (2013) and APRP methods described below, there is no way to separately quantify the three cloud-related terms – the semidirect effect which is considered part of $ERF_{ari}$, the instantaneous component due to changing particle number and size (the Twomey effect), and the subsequent adjustments. Instead, for these two methodologies, the cloud terms are combined into a single term that is included as part of the $ERF_{aci}$:

$$K^C \Delta C = IRF_{aci} + K^C \Delta C_{adjustments} + K^C \Delta C_{semidirect}. \tag{5}$$

$K^C \Delta C$ can be further broken down into contributions from changes in cloud amount and optical properties.

### 2.2.2 Approximate Partial Radiative Perturbation (APRP) Technique

Taylor et al. (2007) developed an efficient method of decomposing perturbations to the TOA SW energy budget from clouds, the cloud-free portion of the atmosphere (here assumed to be dominated by aerosols), and surface albedo. Briefly, the APRP method employs a simple one-layer model of the atmosphere to diagnose the scattering and absorption of SW radiation at the surface and in the atmosphere. This simple model represents the transfer of SW radiation through the atmosphere at every grid point on the globe in terms of a small number of parameters – the insolation, surface albedo, an atmospheric scattering coefficient, and an atmospheric absorptance coefficient. Given the known SW fluxes at the TOA and surface under both clear- and all-sky conditions as well as the total cloud fraction, at each grid point on the globe one can solve for the atmospheric scattering and absorption parameters in this simplified representation such that the upwelling and downwelling SW radiative fluxes at the surface and TOA match those produced by the GCM. Then the sensitivity of TOA albedo to these parameters can be determined, allowing one to isolate the individual contributions from changes in non-cloud atmospheric constituents and from

changes in cloud properties. This technique is an approximation to the more rigorous but difficult-to-implement partial radiative perturbation (PRP) technique, and was shown to closely agree with PRP-derived SW cloud and surface albedo feedbacks (Taylor et al., 2007), with errors that were no larger than 10%. It is uniquely well-suited for quantifying and decomposing aerosol forcing, since aerosols primarily affect scattering and absorption of SW radiation both directly and indirectly through clouds. A more detailed description of APRP and how it is used to estimate the individual aerosol forcing components is provided in Appendix A.2, and code to perform all APRP calculations for this paper is provided in the *Code availability* section.

APRP (denoted below with a superscript $A$) provides estimates of SW ERF (superscript $SW$) that are made up of slightly different term groupings than the IPCC AR6 definition:

$$ERF_{ari}^{A,SW} = IRF_{ari}^{SW} + K^{q,SW}\Delta q = ERF_{ari}^{SW} - K^{C,SW}\Delta C_{semidirect}, \tag{6}$$

and

$$ERF_{aci}^{A,SW} = IRF_{aci}^{SW} + K^{C,SW}\Delta C_{adjustments} + K^{C,SW}\Delta C_{semidirect} = ERF_{aci}^{SW} + K^{C,SW}\Delta C_{semidirect}. \tag{7}$$

The impact of temperature changes on the SW $ERF_{ari}$ can be neglected. Therefore in the SW, APRP's direct effect equals IPCC's direct effect minus the semidirect effect, while APRP's indirect effect equals IPCC's indirect effect plus the semidirect effect. The sum of the direct and indirect SW effects are the same, independent of how the individual components are defined. Thus, from Eqs 6 and 7:

$$ERF_{ari}^{A,SW} + ERF_{aci}^{A,SW} = ERF_{ari}^{SW} + ERF_{aci}^{SW}. \tag{8}$$

Further insight comes from separating $\text{ERF}_{ari}^{A,SW}$ into absorption and scattering components,

$$ERF_{ari}^{A,SW} = ERF_{ari,abs}^{A,SW} + ERF_{ari,scat}^{A,SW}, \tag{9}$$

and from separating $\text{ERF}_{aci}^{A,SW}$ into cloud absorption, amount, and scattering components,

$$ERF_{aci}^{A,SW} = ERF_{aci,abs}^{A,SW} + ERF_{aci,amt}^{A,SW} + ERF_{aci,scat}^{A,SW}. \tag{10}$$

APRP also quantifies the impact of surface albedo changes on TOA SW radiation.

### 2.2.3 LW ERFs: Proxies Derived from Standard Model Output

There is no equivalent to APRP for LW radiation, so instead we compute proxies (denoted with superscript $P$) for LW ERFs using standard model output, following Zelinka et al. (2014) and Smith et al. (2020). The LW direct effect is estimated as the change in clear-sky TOA LW radiation excluding the portion due solely to land surface temperature changes:

$$ERF_{ari}^{P,LW} = \Delta R_{cs}^{LW} - \Delta R_{cs}^{T_0}, \tag{11}$$

where $R^{LW}$ refers to net (downwelling minus upwelling) LW radiation and the subscript $cs$ refers to clear-sky conditions. The change in clear-sky LW radiation due to changes in (land) surface temperature is computed by multiplying the change in surface temperature between the two experiments by the clear-sky surface temperature radiative kernel (Huang et al., 2017):

$$\Delta R_{cs}^{T_0} = K_{cs}^{T_0} \Delta T_0. \tag{12}$$

We compute a proxy for longwave $ERF_{aci}$ as the change in LW cloud radiative effect:

$$ERF_{aci}^{P,LW} = \Delta CRE^{LW} = \Delta R^{LW} - \Delta R_{cs}^{LW}. \tag{13}$$

As shown in Appendix B, our proxy for the LW direct effect equals IPCC's direct effect, minus the semidirect effect, minus masking terms that quantify how much the radiative impact of changes in temperature, humidity, and aerosols are attenuated by the presence of clouds. Our proxy for the LW indirect effect equals IPCC's indirect effect, plus the semidirect effect, plus the aforementioned masking terms.

### 2.2.4 Double Radiation Call Method

Ghan (2013) introduced a method to compute aerosol direct and indirect effects that relies on "aerosol-free" radiative fluxes under both clear- and all-sky conditions. These are produced by performing additional calls to the radiation code during model integration in which all aerosols are neglected. For the models that provide aerosol-free radiative fluxes, we compute the following quantities, which are given the superscript $G$ to indicate Ghan. Ghan (2013) defines the direct forcing as

$$ERF_{ari}^{G} = \Delta R - \Delta R_{af}, \tag{14}$$

the indirect forcing as

$$ERF_{aci}^{G} = \Delta R_{af} - \Delta R_{af,cs}, \tag{15}$$

and a third forcing term as

$$ERF_{other}^{G} = \Delta R_{af,cs}. \tag{16}$$

where the subscript $af$ refers to aerosol-free radiative fluxes and, as above, $\Delta R$ represents the change in net radiation between the control and aerosol-perturbed experiment.

As shown in Appendix B, Ghan's direct aerosol radiative forcing equals IPCC's instantaneous direct forcing minus masking terms that quantify how much the radiative impact of changes in temperature, humidity, surface albedo, and clouds are attenuated by the presence of aerosols. Ghan's indirect effect equals IPCC's indirect effect plus masking terms that quantify how much the radiative impact of changes in temperature, humidity, and surface albedo are attenuated by the presence of clouds under aerosol-free conditions and how much the radiative impact of changes in clouds are attenuated by the presence of aerosols, plus the semidirect effect. Finally, $ERF_{other}^{G}$ – which in the SW Ghan (2013) refers to as the surface albedo forcing – equals $\Delta R_{alb}$ plus the aerosol-free clear-sky radiative contributions from changes in humidity, plus a masking term that quantifies how much the radiative impact of changes in surface albedo are attenuated by the presence of both clouds and aerosols.

## 3 Results

### 3.1 Errors in APRP Implementation of Smith et al. (2020)

We begin by comparing the sum of all APRP-derived SW ERF components with the total change in SW radiation between the perturbed aerosol and control experiment. Since the majority of the results shown below are derived using APRP, we omit the $A$ superscript hereafter. If APRP is correctly implemented, the sum of its components should perfectly reproduce the total change in SW radiation between the control and perturbed experiment. As shown in Figure 1, the APRP implementation in the present study has a negligible residual whereas that in Smith et al. (2020) is generally nonzero, ranging from -0.21 to +0.08 W/m$^2$ across models. These errors, resulting from a mistake in coding as detailed below, are comparable in magnitude to typical values for the total aerosol direct effect and the cloud absorption and amount components of the indirect effect (shown below).

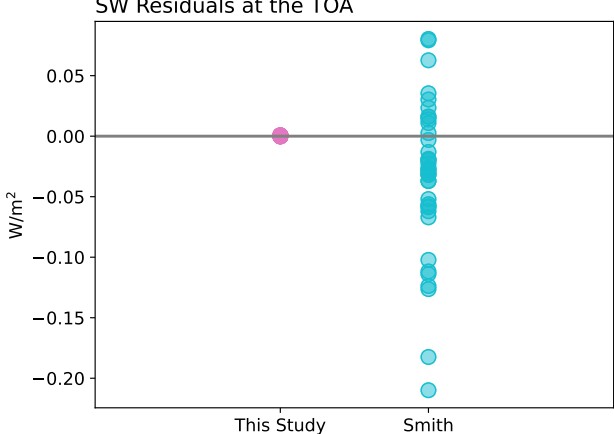

**Figure 1.** Global mean residuals estimated as the difference between the model-produced change in TOA SW fluxes and those estimated by summing the APRP components, shown for the APRP implementation in this study and in Smith et al. (2020). Each marker represents a different CMIP model (all pink markers overlap).

The small residual in Smith's implementation of APRP masks the two much larger errors that are nearly perfectly compensating. The first error – which in isolation leads to overestimated ERF magnitudes – is in how TOA albedo sensitivities are computed, particularly in the calculation of overcast-sky albedo sensitivity to aerosol and cloud absorption and scattering. The code overestimated the impact of changes in scattering or absorption on overcast-sky fluxes because it made use of the raw difference between clear-sky scattering and absorption coefficients. The correct implementation, in contrast, scales this difference by the appropriate factor to account for the fact that changes in the non-cloud portion of a column are attenuated by the presence of clouds in the column. This error is described in further detail in Appendix A.3. The second error – which in isolation leads to underestimated ERF magnitudes – is that net (downwelling minus upwelling) rather than downwelling TOA SW radiation was used in all calculations. This error is described in further detail in Appendix A.4.

To quantify the impact of these two errors, we first corrected both errors in the code of Smith et al. (2020), and verified that it produces results identical to those from the original APRP implementation of Zelinka et al. (2014). The correct implemen-

195 tation is shown as pink bars in Figure 2, and serves as the baseline against which subsequent calculations are compared. We then performed the APRP calculations two more times, once reverting back to the original erroneous insolation formulation, and once reverting back to the original erroneous albedo sensitivity formulation. The ERF estimate derived from the APRP implementation with erroneous insolation is shown by the dotted hatching overlain on the pink bar. This bar has a smaller magnitude than the true value because it uses net (down- minus up-welling radiation) rather than downwelling radiation. The

200 ratio of these two ERF estimates (shown in printed numbers) is 0.77 averaged across models, with a standard deviation of 0.05. This is consistent with the fact that net radiation is equal to the downwelling radiation times (1 minus planetary albedo) and that planetary albedos vary between about 0.2 and 0.3.

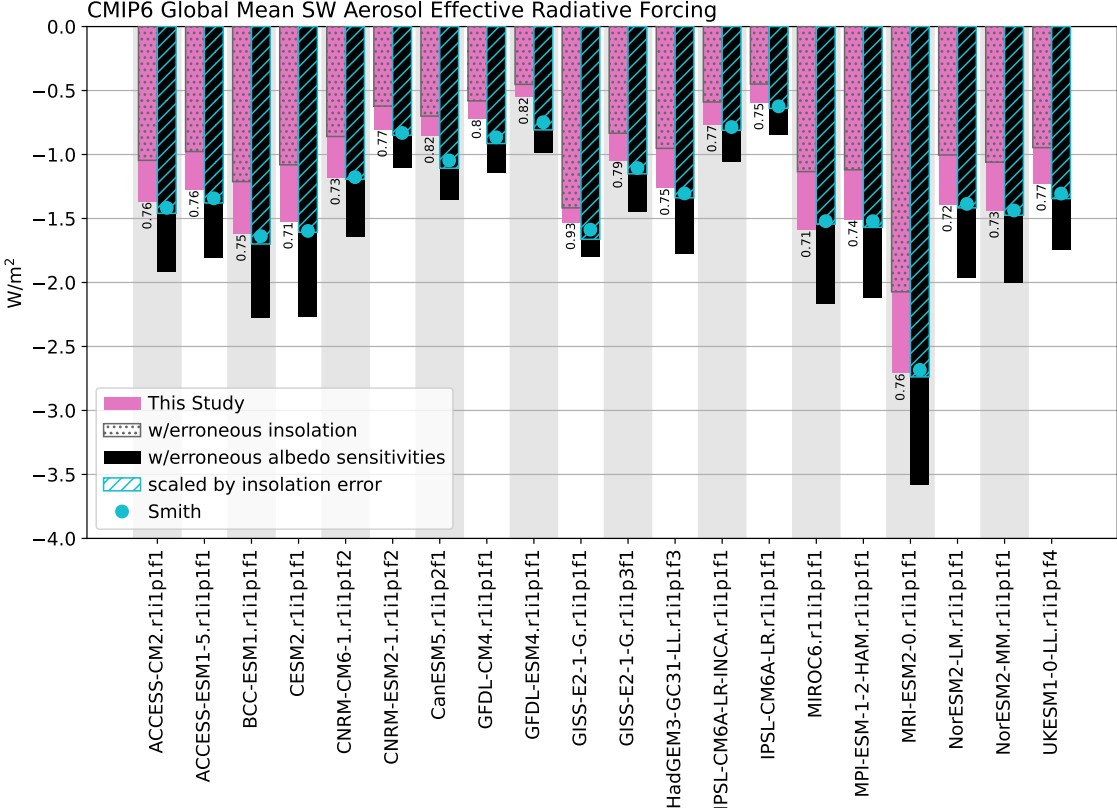

**Figure 2.** Estimates of global mean $\mathrm{ERF}_{ari+aci}^{SW}$ across CMIP6 models. The pink bar shows the values derived in this study using the corrected APRP formulation. Overlain on this with dotted gray hatching is the ERF derived if using net rather than downwelling SW radiation, and the ratio between these two is printed. The adjacent black bar shows the ERF derived if using erroneous albedo sensitivities (but correct insolation). Overlain on this with cyan hatching is this same value scaled by the aforementioned ratio, which closely matches the values derived using the APRP formulation of Smith et al. (2020), which are shown with cyan dots.

The ERF estimates derived from the APRP implementation with erroneous albedo sensitivities (but correct insolation) are shown in the black bars in Figure 2. The erroneous albedo sensitivity formulation leads to ERF values that are biased too large

in magnitude. This arises almost entirely from the scattering components of $\mathrm{ERF}_{aci}^{SW}$ and $\mathrm{ERF}_{ari}^{SW}$, which are overestimated (discussed further below). This occurs because the erroneous code did not properly allow the increase in aerosol scattering to be attenuated by cloud scattering, and did not properly allow the increase in cloud scattering to be attenuated by non-cloud (aerosol) scattering (not shown).

The bar with cyan hatching overlain on the black bar in Figure 2 indicates the result if we scale these overestimated ERFs by

210 the ratios diagnosed above to account for the compensating insolation error. This scaled estimate closely matches the original Smith et al. (2020) formulation in which both errors are present (cyan dots).

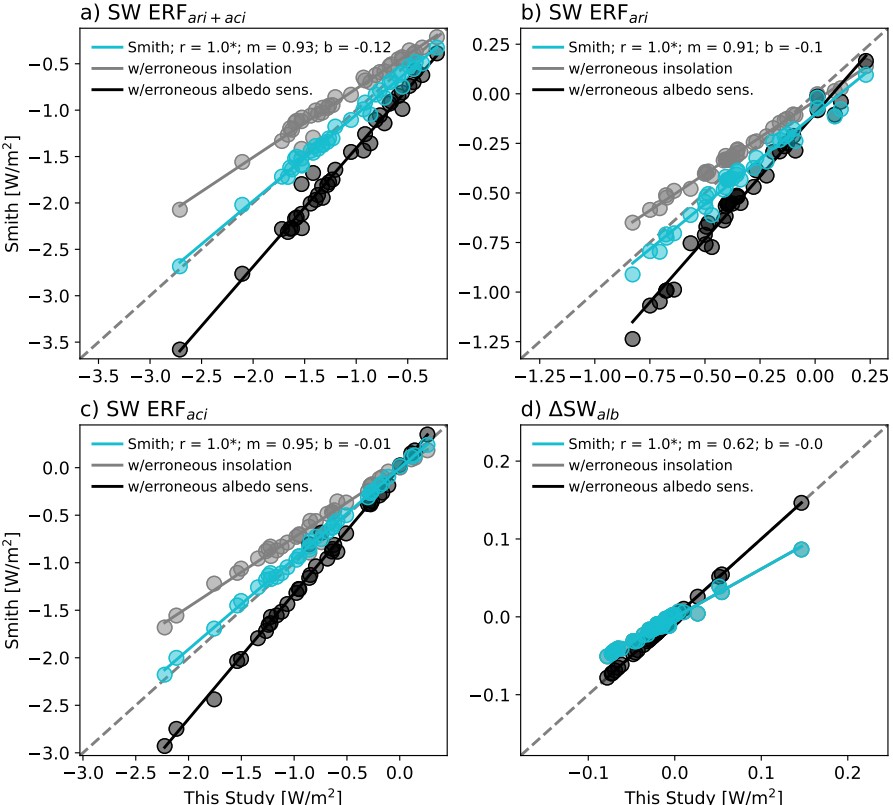

**Figure 3.** Estimates of global mean (a) $\mathrm{ERF}_{ari+aci}^{SW}$, (b) $\mathrm{ERF}_{ari}^{SW}$, (c) $\mathrm{ERF}_{aci}^{SW}$, and (d) $\Delta SW_{alb}$ in each CMIP5 and CMIP6 model derived using the APRP formulation of Smith et al. (2020) scattered against estimates computed in this study. Cyan markers indicate the values estimated by Smith et al. (2020), gray markers indicate values derived using net rather than downwelling SW radiation (but correct albedo sensitivities), and black markers indicate values derived using erroneous albedo sensitivities (but correct insolation). Gray markers in panel d are overlain by cyan markers. Correlation, slope, and intercept are reported as r, m, and b, respectively, and correlations that are statistically significant at 95% confidence are indicated with an asterisk.

Hence the two errors are individually substantial in magnitude, but act in opposite directions that almost perfectly compensate, leading to total ERF values that are in good agreement with the correct values estimated here using the APRP implementation of Zelinka et al. (2014). This also holds for the geographic structure of the ERF components, which are highly consistent

between the two APRP implementations but differ quantitatively (not shown). This appears to be a fortuitous result. Moreover, while this good agreement with the correct values holds for the total ERF, several of its sub-components show larger biases, as discussed next.

In Figure 3 we scatter estimates of $\mathrm{ERF}^{SW}_{ari+aci}$, its two sub-components ($\mathrm{ERF}^{SW}_{ari}$ and $\mathrm{ERF}^{SW}_{aci}$), and the $\Delta SW_{alb}$ term derived using the APRP implementation of Smith et al. (2020) against those derived using the implementation of Zelinka

et al. (2014). $\mathrm{ERF}^{SW}_{ari}$ and $\mathrm{ERF}^{SW}_{aci}$ are further broken down into their scattering and absorption sub-components in Figure 4. The aforementioned compensating biases are indicated by the gray and black markers, which show the ERF values derived if correcting each bias in turn. Correcting the albedo sensitivity formulation but keeping the insolation bias leads to ERF values that are too small in magnitude (Figure 3-4, gray markers). This is expected because a given change in albedo will produce a weaker TOA impact when using a smaller (downwelling minus upwelling) SW radiation stream. Further details are provided

in Appendix A.4.

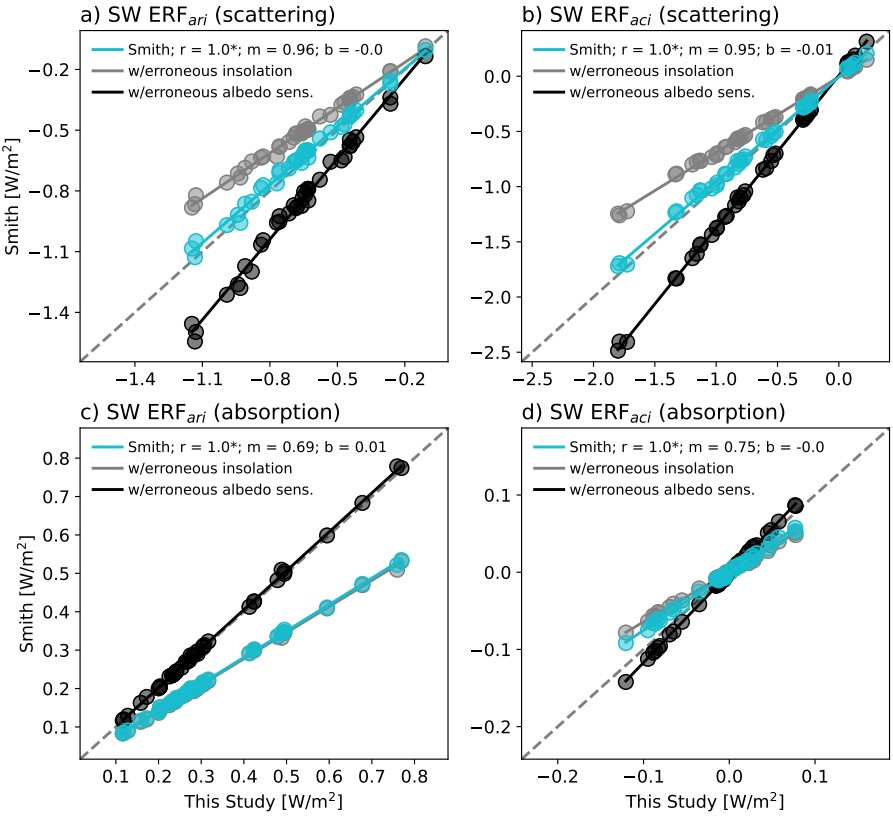

**Figure 4.** As in Figure 3, but for the scattering and absorption sub-components of $\mathrm{ERF}^{SW}_{ari}$ and $\mathrm{ERF}^{SW}_{aci}$.

Correcting the insolation but keeping the erroneous albedo sensitivity formulation (black markers) has the opposite effect: In most cases it leads to ERF values that are too large in magnitude for reasons described in Appendix A.3. This indicates that using correct values of downwelling (rather than net) SW radiation exposes the erroneous albedo sensitivity formulation wherein changes in scattering or absorption have too-strong an effect on TOA SW radiation. This error primarily manifests itself

in the scattering components of $\text{ERF}_{ari}^{SW}$ and $\text{ERF}_{aci}^{SW}$ (Figure 4a and b, black markers), whereas the absorption components are largely corrected when the insolation is corrected (Figure 4c and d, black markers). The albedo sensitivity formulation error has a larger impact on the scattering components because clouds scatter more SW radiation than they absorb. This means that neglecting the attenuation of the change in aerosol scattering by cloud scattering induces a larger error than neglecting cloud absorption when computing the impact of a change in aerosol absorption. Finally, the $\Delta SW_{alb}$ bias vanishes once the insolation

bias is corrected (Figure 3d, black markers). This is because, unlike for the sensitivity of planetary albedo to atmospheric scattering and absorption parameters, the sensitivity of planetary albedo to surface albedo is implemented correctly in Smith et al. (2020).

For the ERF components that depend only on scattering of radiation, these two biases largely compensate such that the values reported in Smith et al. (2020) agree well with those estimated here (Figure 4a and b, cyan markers). However, for the

240 absorption components of $\text{ERF}_{ari}^{SW}$ and $\text{ERF}_{aci}^{SW}$ (for which the albedo sensitivity formulation was not as biased) and for the surface albedo term (which does not depend on how the atmospheric scattering and absorption sensitivities are formulated), the insolation error is largely uncompensated. Therefore, the values reported in Smith et al. (2020) remain systematically biased for these components. This is most apparent for $\Delta SW_{alb}$ (Figure 3d) and the absorption component of $\text{ERF}_{ari}^{SW}$ (Figure 4c).

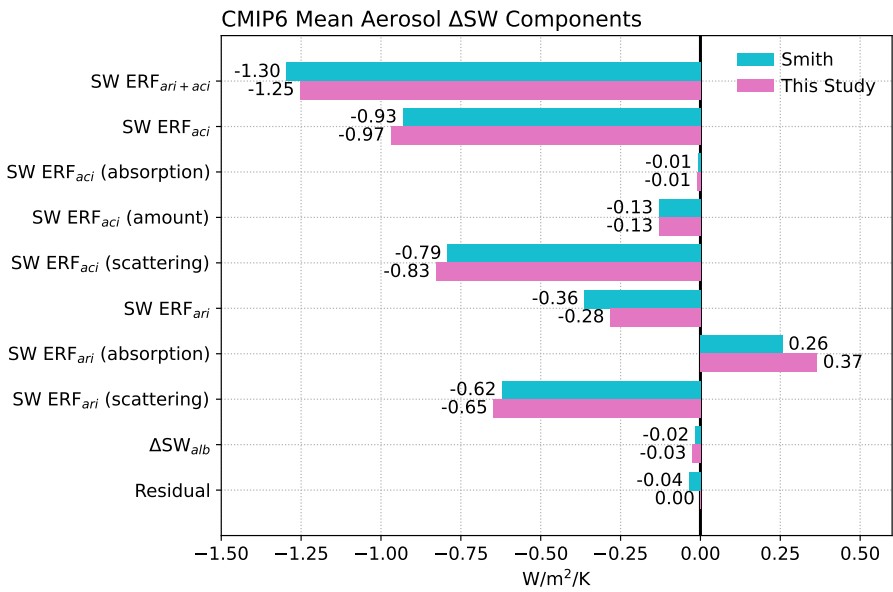

**Figure 5.** Components of the change in global mean SW TOA radiation averaged across CMIP6 models as computed by Smith et al. (2020) and as computed in this study.

Figure 5 shows the CMIP6 multi-model mean SW ERF components estimated using the APRP formulation of Smith et al. (2020) and as derived in this study. Note that the Smith et al. values differ slightly from those reported in that study because here we include four additional CMIP6 models not assessed in that study and exclude EC-Earth3 which was used in that study. We find that all but three ERF values are within 5% of those diagnosed by Smith et al. (2020): The very small $\Delta SW_{alb}$ component increases in strength from -0.02 to -0.03 W/m$^2$. More importantly, the negative $\mathrm{ERF}_{ari}^{SW}$ diagnosed here is roughly 20% weaker in magnitude. This is primarily because the positive absorbing component of $\mathrm{ERF}_{ari}^{SW}$ is roughly 40% larger in the present study.

## 3.2 Validation of ERF Estimates Against the Double-Call Method

In Figure 6 we compare the SW ERF values estimated using APRP with those derived using the technique of Ghan (2013) in which radiation calculations are performed an additional time with all aerosols neglected. This comparison is done for 10 models (indicated with asterisks in Table 2) that provided the necessary aerosol-free diagnostics. As mentioned earlier and as detailed in Appendix B, APRP and Ghan (2013) define each of these terms differently, so differences between the two estimates do not necessarily indicate errors. However, one would expect these to be closely related and, when implemented correctly, APRP values closely reproduce the double-call values. Indeed, both APRP implementations are well correlated with the double-call method for each ERF component, but the RMSD values are much smaller when using the values derived in this study.

Even when correctly implemented, APRP-estimated $\mathrm{ERF}_{ari}^{SW}$ values are biased towards stronger negative / weaker positive values than those derived via double-call (Figure 6b). This is compensated somewhat by the $\mathrm{ERF}_{aci}^{SW}$ values, for which the APRP method is biased towards smaller negative values (Figure 6c), as also shown for a single model in Zelinka et al. (2014). APRP also yields $\Delta SW_{alb}$ values that are weaker in magnitude than produced by the double-call (Figure 6d). As shown in Appendix B, these differences are expected because the two methods are diagnosing slightly different things for each of the individual components, with the offset arising from small masking terms. For example, Ghan (2013) defines $\Delta SW_{alb}$ as the change in clear-sky aerosol-free SW fluxes (Eq 16) whereas the APRP quantifies this as the change in all-sky radiation due to changes in surface albedo. Hence the latter calculation allows the radiative impact of changing surface albedo to be attenuated by the presence of clouds and aerosols (i.e., masking effects) and avoids aliasing in contributions from humidity changes that impact SW absorption. Indeed, the double-call $\Delta SW_{alb}$ values agree closely with the clear-sky surface albedo component diagnosed by APRP scaled by the clear-sky fraction (r=1.0; RMSD=0.02; not shown).

The multi-model mean maps show excellent agreement in the spatial structure of each component between the correct implementation of APRP and Ghan (2013) methodologies, albeit with quantitative differences (Figure 7). Both methods agree on negative $\mathrm{ERF}_{ari+aci}^{SW}$ over the vast majority of the Northern Hemisphere, with largest negative values over the Northern Indian Ocean and over Southeast Asia. Negative values are also present across the North Pacific and North Atlantic Oceans and over most of the Northern Hemisphere continental regions, and local maxima are present just west of South America and Africa.

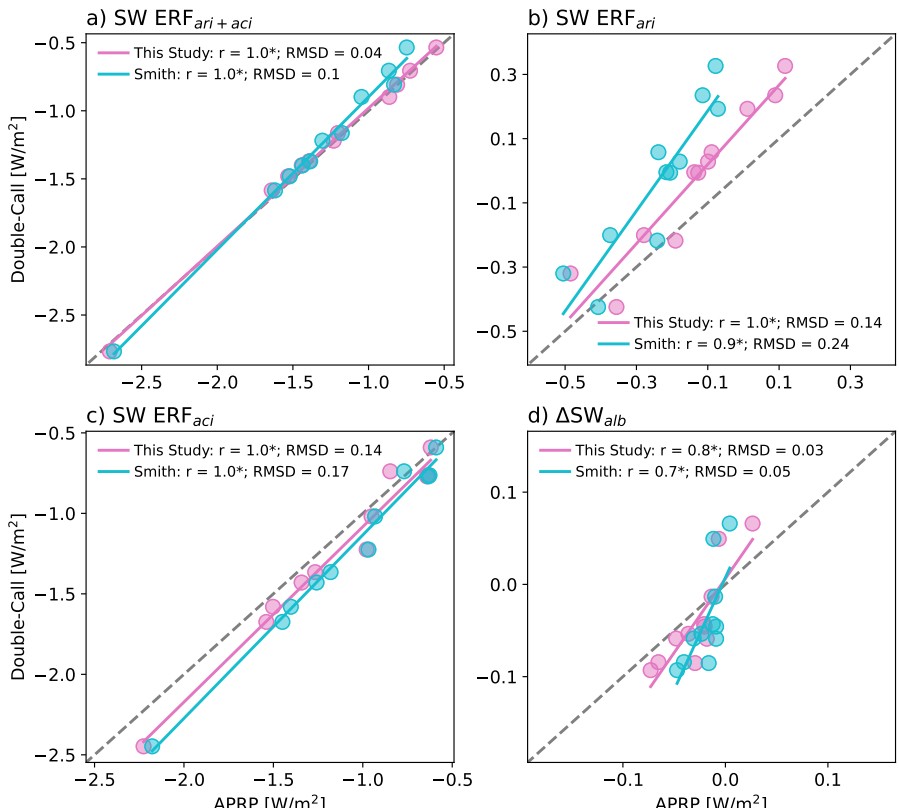

**Figure 6.** Estimates of global mean (a) $\mathrm{ERF}^{SW}_{ari+aci}$, (b) $\mathrm{ERF}^{SW}_{ari}$, (c) $\mathrm{ERF}^{SW}_{aci}$, and (d) $\Delta SW_{alb}$ derived using the APRP method as implemented in this study (pink) and in Smith et al. (2020) (blue) in each CMIP6 model scattered against estimates computed using the double-call method of Ghan (2013). Note that we are referring to Ghan's $\mathrm{ERF}^{SW}_{other}$ as $\Delta SW_{alb}$ here even though the former receives contributions from more than just changes in surface albedo (see Appendix B). The correlation and root mean square difference are reported as r and RMSD, respectively, and correlations that are statistically significant at 95% confidence are indicated with an asterisk.

Both methods diagnose a positive $\mathrm{ERF}^{SW}_{ari}$ over Africa (Figure 7, row 2). The APRP results indicate that this is a region of strong aerosol absorption that is not fully countered by aerosol scattering (not shown), perhaps indicating a role of black carbon from biomass burning. Just east of this, the two methods agree on a negative direct effect over the northern Indian
Ocean. Despite the large aerosol emission sources, the $\mathrm{ERF}^{SW}_{ari}$ is not particularly large over Southeast Asia because of close compensation between the absorbing and scattering components (not shown). The spatial structure of the total ERF is very consistent with the $\mathrm{ERF}^{SW}_{aci}$ map (Figure 7, row 3), highlighting the dominant role played by the indirect effect in models' total aerosol forcing. Weak negative $\Delta SW_{alb}$ values are present over much of the NH continents, likely due to increased snow cover in response to aerosol-induced cooling (Figure 7, row 4). An exception is a region of large positive $\Delta SW_{alb}$ over the
Himalayas, which may result from black carbon deposition on snow.

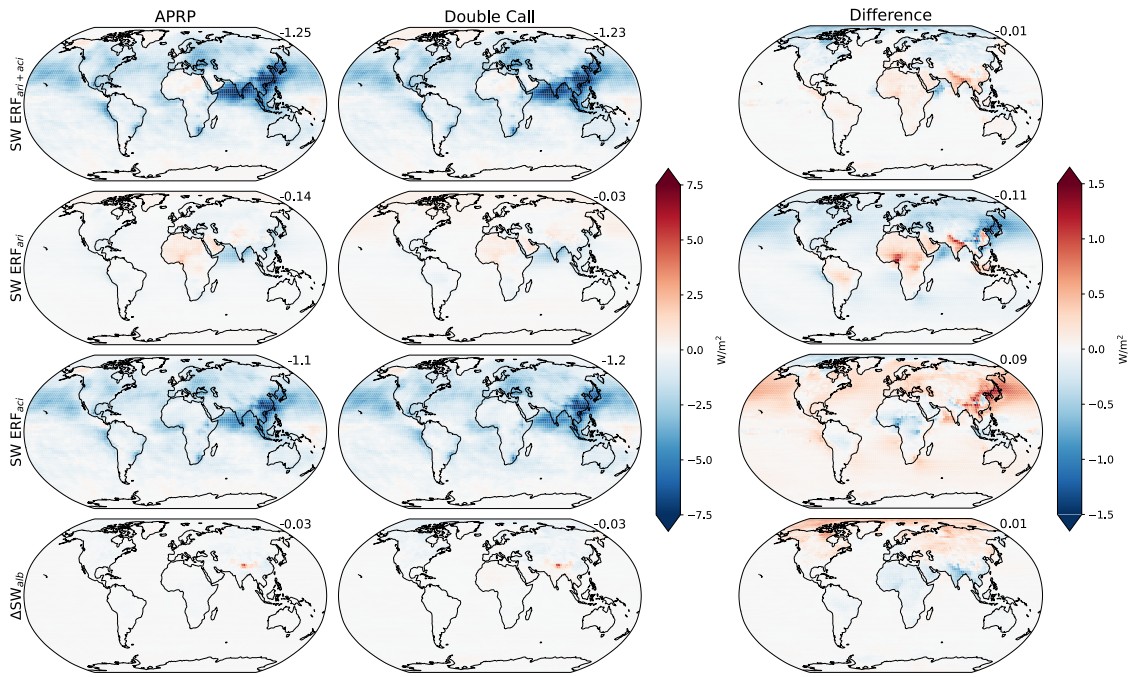

**Figure 7.** $ERF^{SW}_{ari+aci}$ (row 1), $ERF^{SW}_{ari}$ (row 2), $ERF^{SW}_{aci}$ (row 3), and $\Delta SW_{alb}$ (row 4) estimated by the APRP method (column 1), the double-call method (column 2), and their difference (column 3). Area-weighted global mean values are printed in the top right corner of each panel. Results are averaged across all models providing double-call output.

Consistent with the comparison for a single model shown in Zelinka et al. (2014), locations where APRP overestimates the negative $ERF^{SW}_{ari}$ are co-located with locations where it underestimates $ERF^{SW}_{aci}$, and vice versa. This is noticeable over the North Pacific Ocean downwind of the Eastern Asia $ERF^{SW}_{ari+aci}$ maximum. These opposite-signed differences with respect to the double-call method cancel such that the total $ERF^{SW}_{ari+aci}$ maps are nearly identical (Figure 7, row 1). The global mean

root-mean-square difference between APRP and double-call estimates of $ERF^{SW}_{ari+aci}$ is less than 10% of the standard deviation of either field, confirming that both methods agree as to spatial variations.

Longwave ERF components cannot be derived using APRP, but we can derive proxies for the direct and indirect components from standard model output (Section 2.2.3). Specifically, we use the change in clear-sky LW radiation with the contribution from changes in surface temperature removed as a proxy for the LW direct effect. Recall that the formulation of Ghan (2013)

defines the direct effect as the difference between changes in LW radiation under aerosol-free conditions and under all-sky conditions (Eq 14). As demonstrated mathematically in Appendix B and as shown in Figure 8a, our $ERF^{LW}_{ari}$ proxy is closely related to the sum of Ghan's $ERF^{LW,G}_{ari}$ and $ERF^{LW,G}_{other}$ terms with the contribution from changes in surface temperature removed. In other words, the change in clear-sky LW radiation is made up of the direct aerosol effect as defined in Ghan (2013) plus perturbations due to aerosol-induced fast adjustments of humidity and temperature, including at the surface.

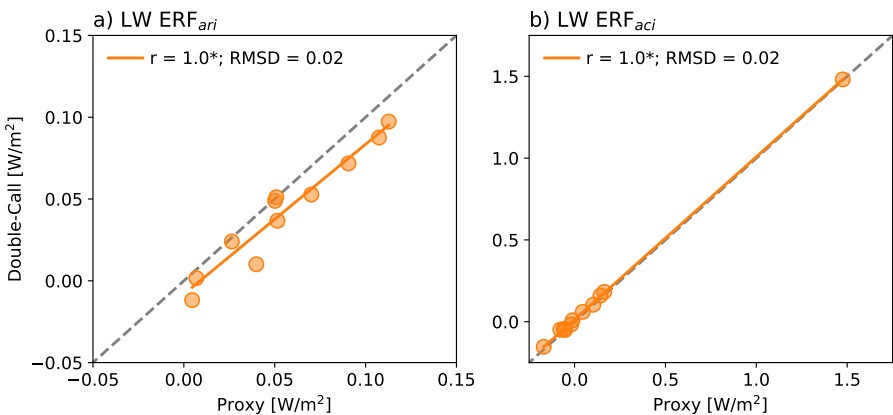

**Figure 8.** Estimates of global mean (a) $\text{ERF}_{ari}^{LW}$ and (b) $\text{ERF}_{aci}^{LW}$ derived using standard output from each CMIP6 model scattered against estimates computed using the double-call method of Ghan (2013). The double-call estimate in (a) is the sum of $\text{ERF}_{ari}^{G,LW}$ and $\text{ERF}_{other}^{G,LW}$ with the surface temperature response removed (see Appendix B).

The change in LW CRE – our proxy for the LW indirect effect – is very well-correlated with $\text{ERF}_{aci}^{LW}$ derived with the double-call method (Figure 8b), justifying its use as a proxy for the LW indirect effect. The model with large $\text{ERF}_{aci}^{LW}$ apparent in Figure 8b is MRI-ESM2-0, whose CMIP5-era counterpart also exhibited a large indirect component in the LW. This model was among only a few in CMIP5 that parameterized aerosol impacts on ice clouds, and it exhibited large changes in the amount and optical depth of high clouds in response to aerosols (Zelinka et al., 2014). This remains the model with largest

aerosol effects on high clouds and therefore on LW cloud-radiative fluxes, as discussed further in Smith et al. (2020).

   We conclude from this section that the APRP technique yields aerosol ERF values that agree well with the independent double-call method, both in the global average and in the spatial distribution. Given that it does not require advanced diagnostics that may not be available in many models and experiments, it is an attractive method for systematically diagnosing these values across a broad suite of climate models. The typically smaller LW components of aerosol forcing are likewise well-captured

using simpler diagnostics that are widely available.

### 3.3 Summary of Corrected APRP-Derived ERF Values

Having established that biases were present in the values of aerosol ERF provided in Smith et al. (2020) and that the values derived herein have negligible residuals and are in better agreement with the double-call method, we now provide the aerosol ERF values for all available models, along with their breakdown into components (Tables 1-2). This includes several CMIP6 models

that were not available at the time Smith et al. (2020) was published. Although our APRP implementation is unchanged from that used in Zelinka et al. (2014), we report here CMIP5 results that include three additional models (bcc-csm1-1, FGOALS-s2, and MPI-ESM-LR) that were not included in the earlier paper.

**Table 1.** Aerosol ERF$_{ari}$, ERF$_{aci}$, and ERF$_{ari+aci}$ values [W/m$^2$] estimated for individual CMIP5 models, separated into SW, LW, and net components. The ERF$_{ari}^{SW}$ is further separated into scattering and absorption components, and the ERF$_{aci}^{SW}$ is further separated into scattering, absorption, and amount components. Also shown are the multi-model means and across-model standard deviations, both computed using only one ensemble member per model. Forcings are for present-day (year 2000) relative to pre-industrial conditions.

| Model | SW | | | | | | | | LW | | | Net | | |
| --- | --- | --- | --- | --- | --- | --- | --- | --- | --- | --- | --- | --- | --- | --- |
| | ARI | | | ACI | | | | | | | | | | |
| | scat | abs | sum | scat | abs | amt | sum | ARI+ACI | ARI | ACI | ARI+ACI | ARI | ACI | ARI+ACI |
| CSIRO-Mk3-6-0.r1i1p1 | -1.13 | 0.49 | -0.64 | -0.76 | 0.08 | 0.00 | -0.68 | -1.33 | -0.06 | -0.21 | -0.27 | -0.70 | -0.89 | -1.59 |
| CanESM2.r1i1p1 | -0.58 | 0.17 | -0.41 | -0.52 | 0.01 | -0.01 | -0.51 | -0.91 | 0.06 | -0.04 | 0.02 | -0.35 | -0.55 | -0.90 |
| FGOALS-s2.r1i1p1 | -0.77 | 0.20 | -0.57 | 0.11 | 0.01 | 0.02 | 0.14 | -0.42 | 0.05 | -0.06 | -0.01 | -0.52 | 0.08 | -0.44 |
| GFDL-CM3.r1i1p1 | -0.91 | 0.41 | -0.50 | -1.00 | -0.04 | -0.13 | -1.17 | -1.66 | 0.02 | 0.02 | 0.04 | -0.48 | -1.15 | -1.63 |
| HadGEM2-A.r1i1p1 | -0.53 | 0.26 | -0.27 | -0.99 | 0.06 | -0.13 | -1.06 | -1.33 | 0.07 | -0.05 | 0.02 | -0.20 | -1.11 | -1.31 |
| IPSL-CM5A-LR.r1i1p1 | -0.66 | 0.28 | -0.38 | -0.23 | -0.01 | 0.13 | -0.11 | -0.48 | -0.05 | -0.22 | -0.27 | -0.43 | -0.33 | -0.76 |
| MIROC5.r1i1p1 | -0.66 | 0.16 | -0.50 | -0.93 | -0.01 | -0.28 | -1.22 | -1.72 | 0.13 | 0.27 | 0.40 | -0.37 | -0.95 | -1.32 |
| MPI-ESM-LR.r1i1p1 | -0.71 | 0.49 | -0.22 | 0.10 | -0.08 | -0.01 | 0.00 | -0.22 | -0.07 | -0.10 | -0.17 | -0.29 | -0.10 | -0.39 |
| MPI-ESM-LR.r1i1p2 | -1.15 | 0.68 | -0.47 | 0.13 | -0.07 | 0.04 | 0.11 | -0.36 | -0.12 | -0.17 | -0.29 | -0.59 | -0.06 | -0.65 |
| MRI-CGCM3.r1i1p1 | -0.11 | 0.12 | 0.01 | -1.79 | -0.09 | -0.23 | -2.12 | -2.11 | 0.03 | 0.95 | 0.98 | 0.04 | -1.17 | -1.13 |
| NorESM1-M.r1i1p1 | -0.63 | 0.29 | -0.34 | -0.80 | 0.03 | 0.18 | -0.59 | -0.93 | 0.04 | -0.17 | -0.13 | -0.30 | -0.76 | -1.06 |
| bcc-csm1-1.r1i1p1 | -0.93 | 0.23 | -0.71 | 0.23 | 0.01 | 0.03 | 0.26 | -0.44 | -0.01 | -0.05 | -0.06 | -0.72 | 0.21 | -0.51 |
| Mean | -0.69 | 0.28 | -0.41 | -0.60 | -0.00 | -0.04 | -0.64 | -1.05 | 0.02 | 0.03 | 0.05 | -0.39 | -0.61 | -1.00 |
| 1-$\sigma$ | 0.25 | 0.12 | 0.20 | 0.58 | 0.05 | 0.14 | 0.68 | 0.60 | 0.06 | 0.32 | 0.34 | 0.21 | 0.48 | 0.42 |

Every CMIP6 model agrees on a negative ERF$_{ari}^{SW}$ due to scattering and a positive ERF$_{ari}^{SW}$ due to absorption, but the relative strengths vary, leading to a lack of agreement on the sign of ERF$_{ari}^{SW}$. All CMIP6 models have negative ERF$_{aci}^{SW}$ values. This is due to the dominance of strong negative ERF$_{aci}^{SW}$ scattering components, with two exceptions: The r1i1p1f1 and r1i1p1f2 variants of the GISS-E2-1-G model have small positive ERF$_{aci}^{SW}$ cloud scattering components, but anomalously strong negative ERF$_{aci}^{SW}$ cloud amount components. This is consistent with the fact that the 'p1' physics variants of GISS-E2-1-G parameterize aerosol effects on clouds by directly relating anthropogenic aerosol mass to a change in (only) total cloud cover (Miller et al., 2021). In the 'p3' physics variants, in contrast, aerosols act as cloud condensation nuclei and change (only) cloud optical depth; hence the r1i1p3f1 variant of GISS-E2-1-G has a moderate negative ERF$_{aci}^{SW}$ cloud scattering component and weak ERF$_{aci}^{SW}$ cloud amount component (Table 2). In all models, ERF$_{ari+aci}^{SW}$ is negative.

ERF$_{ari}^{LW}$ is positive in nearly all CMIP6 models but is small with opposite sign relative to its SW counterpart. ERF$_{aci}^{LW}$ values are generally small except for the models identified in Smith et al. (2020) as parameterizing aerosol effects on ice clouds (CESM2, MIROC6, MRI-ESM2-0 and NorESM2-LM), for which the ERF$_{aci}^{LW}$ values are non-negligible and positive. Finally,

**Table 2.** As in Table 1, but for CMIP6 models. Models that provided aerosol-free diagnostics (allowing for the comparisons in Figures 5-7) are indicated with asterisks. Multi-model means and across-model standard deviations are computed using only one ensemble member per model, but we treat the r1i1p1f1 and r1i1p3f1 members of GISS-E2-1-G as separate models. Forcings are for present-day (year 2014) relative to pre-industrial conditions.

| Model | SW | | | | | | | | LW | | | Net | | |
| --- | --- | --- | --- | --- | --- | --- | --- | --- | --- | --- | --- | --- | --- | --- |
| | ARI | | | ACI | | | | | | | | | | |
| | scat | abs | sum | scat | abs | amt | sum | ARI+ACI | ARI | ACI | ARI+ACI | ARI | ACI | ARI+ACI |
| ACCESS-CM2.r1i1p1f1 | -0.84 | 0.42 | -0.42 | -0.82 | -0.01 | -0.13 | -0.96 | -1.37 | 0.17 | 0.04 | 0.21 | -0.25 | -0.92 | -1.17 |
| ACCESS-ESM1-5.r1i1p1f1 | -0.44 | 0.28 | -0.16 | -1.04 | 0.05 | -0.13 | -1.12 | -1.28 | 0.09 | -0.02 | 0.07 | -0.07 | -1.14 | -1.21 |
| BCC-ESM1.r1i1p1f1 | -1.13 | 0.30 | -0.83 | -0.62 | -0.09 | -0.09 | -0.79 | -1.62 | 0.03 | 0.06 | 0.09 | -0.80 | -0.73 | -1.53 |
| CESM2.r1i1p1f1 | -0.26 | 0.50 | 0.23 | -1.80 | 0.04 | -0.00 | -1.76 | -1.53 | 0.09 | 0.10 | 0.19 | 0.32 | -1.66 | -1.34 |
| CNRM-CM6-1.r1i1p1f2 | -0.65 | 0.29 | -0.36 | -0.83 | -0.07 | 0.05 | -0.85 | -1.20 | 0.05 | -0.05 | 0.00 | -0.31 | -0.90 | -1.21 |
| CNRM-ESM2-1.r1i1p1f2 | -0.44 | 0.25 | -0.19 | -0.53 | -0.06 | -0.03 | -0.62 | -0.81 | 0.03 | -0.02 | 0.01 | -0.16 | -0.64 | -0.80 |
| CanESM5.r1i1p2f1 | -0.64 | 0.76 | 0.12 | -0.91 | 0.08 | -0.14 | -0.98 | -0.86 | 0.04 | -0.08 | -0.04 | 0.16 | -1.06 | -0.90 |
| GFDL-CM4.r1i1p1f1 | -0.68 | 0.59 | -0.09 | -0.55 | 0.00 | -0.09 | -0.64 | -0.73 | 0.05 | -0.06 | -0.01 | -0.04 | -0.70 | -0.74 |
| GFDL-ESM4.r1i1p1f1 | -0.68 | 0.77 | 0.09 | -0.59 | 0.01 | -0.06 | -0.64 | -0.55 | 0.00 | -0.17 | -0.17 | 0.09 | -0.81 | -0.72 |
| GISS-E2-1-G.r1i1p1f1 | -0.94 | 0.27 | -0.67 | 0.07 | 0.02 | -0.95 | -0.86 | -1.53 | 0.12 | 0.10 | 0.22 | -0.55 | -0.76 | -1.31 |
| GISS-E2-1-G.r1i1p1f2 | -0.88 | 0.20 | -0.68 | 0.07 | 0.03 | -0.84 | -0.74 | -1.42 | 0.06 | 0.10 | 0.16 | -0.62 | -0.64 | -1.26 |
| GISS-E2-1-G.r1i1p3f1 | -0.99 | 0.24 | -0.75 | -0.23 | -0.01 | -0.07 | -0.30 | -1.05 | 0.12 | -0.05 | 0.07 | -0.63 | -0.35 | -0.98 |
| HadGEM3-GC31-LL.r1i1p1f3 | -0.83 | 0.42 | -0.41 | -0.78 | -0.01 | -0.07 | -0.86 | -1.26 | 0.12 | -0.00 | 0.12 | -0.29 | -0.86 | -1.15 |
| IPSL-CM6A-LR-INCA.r1i1p1f1 | -0.69 | 0.20 | -0.49 | -0.29 | -0.01 | 0.03 | -0.28 | -0.77 | 0.02 | -0.07 | -0.05 | -0.47 | -0.35 | -0.82 |
| IPSL-CM6A-LR.r1i1p1f1 | -0.63 | 0.23 | -0.40 | -0.27 | -0.01 | 0.09 | -0.19 | -0.59 | 0.00 | -0.07 | -0.07 | -0.40 | -0.26 | -0.66 |
| IPSL-CM6A-LR.r2i1p1f1 | -0.63 | 0.23 | -0.40 | -0.28 | -0.01 | 0.02 | -0.28 | -0.68 | -0.03 | -0.08 | -0.11 | -0.43 | -0.36 | -0.79 |
| IPSL-CM6A-LR.r3i1p1f1 | -0.63 | 0.24 | -0.39 | -0.25 | -0.02 | 0.10 | -0.16 | -0.56 | -0.02 | -0.05 | -0.07 | -0.41 | -0.21 | -0.62 |
| IPSL-CM6A-LR.r4i1p1f1 | -0.63 | 0.23 | -0.40 | -0.28 | -0.01 | 0.03 | -0.26 | -0.66 | 0.00 | -0.07 | -0.07 | -0.40 | -0.33 | -0.73 |
| MIROC6.r11i1p1f1 | -0.47 | 0.12 | -0.35 | -1.13 | -0.09 | -0.03 | -1.24 | -1.59 | 0.11 | 0.41 | 0.52 | -0.24 | -0.83 | -1.07 |
| MIROC6.r1i1p1f1 | -0.48 | 0.13 | -0.35 | -1.13 | -0.08 | -0.01 | -1.22 | -1.58 | 0.10 | 0.36 | 0.46 | -0.25 | -0.86 | -1.11 |
| MPI-ESM-1-2-HAM.r1i1p1f1 | -0.26 | 0.28 | 0.01 | -1.33 | 0.03 | -0.24 | -1.54 | -1.53 | 0.05 | 0.10 | 0.15 | 0.06 | -1.44 | -1.38 |
| MRI-ESM2-0.r1i1p1f1 | -0.76 | 0.27 | -0.48 | -1.73 | -0.12 | -0.38 | -2.23 | -2.71 | 0.01 | 1.48 | 1.49 | -0.47 | -0.75 | -1.22 |
| NorESM2-LM.r1i1p1f1 | -0.43 | 0.30 | -0.13 | -1.16 | -0.00 | -0.10 | -1.27 | -1.39 | 0.07 | 0.14 | 0.21 | -0.06 | -1.13 | -1.19 |
| NorESM2-LM.r1i1p2f1 | -0.45 | 0.31 | -0.14 | -1.32 | 0.01 | -0.19 | -1.50 | -1.64 | 0.09 | 0.16 | 0.25 | -0.05 | -1.34 | -1.39 |
| NorESM2-MM.r1i1p1f1 | -0.42 | 0.32 | -0.10 | -1.19 | 0.02 | -0.17 | -1.34 | -1.44 | 0.11 | 0.04 | 0.15 | 0.01 | -1.30 | -1.29 |
| UKESM1-0-LL.r1i1p1f4 | -0.76 | 0.48 | -0.28 | -0.85 | -0.01 | -0.10 | -0.95 | -1.23 | 0.11 | -0.01 | 0.10 | -0.17 | -0.96 | -1.13 |
| Mean | -0.65 | 0.37 | -0.28 | -0.83 | -0.01 | -0.13 | -0.97 | -1.25 | 0.07 | 0.09 | 0.16 | -0.21 | -0.88 | -1.09 |
| 1-$\sigma$ | 0.23 | 0.17 | 0.28 | 0.47 | 0.05 | 0.21 | 0.49 | 0.47 | 0.05 | 0.34 | 0.33 | 0.28 | 0.34 | 0.24 |

the net (LW+SW) ERF$_{ari+aci}$ is systematically negative across all models, primarily due to the systematically negative indirect component that is generally larger in magnitude than the direct component, which is small or also negative.

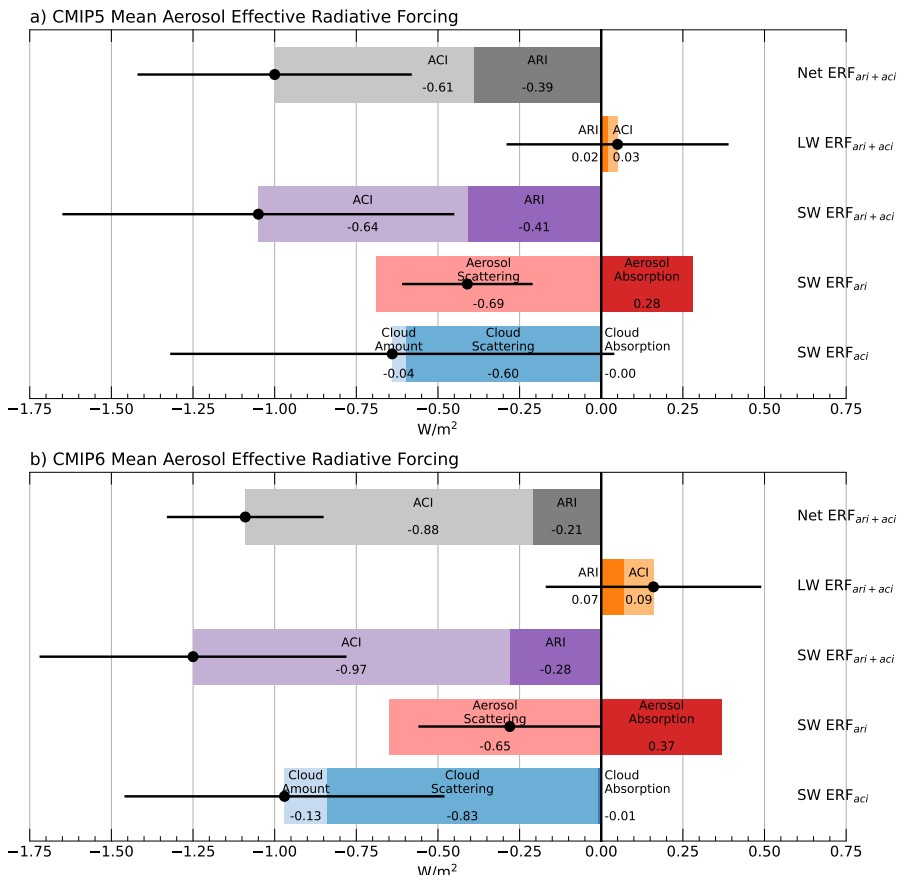

**Figure 9.** Global mean ERF$_{ari+aci}$ values averaged across (a) CMIP5 and (b) CMIP6 models, separated into ERF$_{ari}$ and ERF$_{aci}$ for LW, SW, and net (LW+SW) radiation. ERF$_{ari}^{SW}$ is further separated into its scattering and absorption components. ERF$_{aci}^{SW}$, is further separated into its amount, scattering, and absorption components. The sum of terms in each row is indicated by the black dot, with the inter-model standard deviation of each sum indicated by the horizontal error bar.

To aid in visualizing the model results and to connect back to Figure 10 in Zelinka et al. (2014) and Figure 10 in Smith et al. (2020), multi-model mean ERF values are shown for both the CMIP5 and CMIP6 models in Figure 9. We strongly caution against over-interpreting differences between the two ensembles for two reasons: First, they differ in how "present-day" is defined (2000 for CMIP5 and 2014 for CMIP6). Second, the fraction of models that represent aerosol effects on cloud optical properties and/or lifetimes differs, with three CMIP5 models (FGOALS-s2, MPI-ESM-LR, and bcc-csm1-1) lacking any representation of indirect effects, some models representing only the first indirect effect, and some representing both the first and second indirect effects (Rotstayn et al., 2013; Wilcox et al., 2013). With these caveats in mind, we note that the negative net (LW+SW) ERF$_{aci}$ is nearly 50% stronger in magnitude in CMIP6, while the smaller net ERF$_{ari}$ is roughly halved in CMIP6. Close compensation between these two changes mean that the overall net ERF$_{ari+aci}$ is only slightly larger in

CMIP6. The negative ERF$_{aci}$ is stronger in CMIP6 due primarily to stronger SW cooling from a greater sensitivity of cloud scattering and amount to aerosols (partly related to the fact that three models in CMIP5 do not incorporate indirect effects). The negative ERF$_{ari}$ is weaker in CMIP6 due primarily to stronger SW heating from absorbing aerosols. These same conclusions hold even if we compare CMIP5 and CMIP6 averages based only on models from centers that contributed to both phases (not shown).

## 4 Conclusions

Accurate values of radiative forcings across a broad suite of climate models is a prerequisite for proper understanding of the drivers of inter-model differences in climate response. Model-to-model differences in aerosol radiative forcing are particularly large and come from both aerosol direct and indirect components, each of which has competing contributions from changes in scattering and absorption of SW radiation. In this study we corrected estimates of aerosol effective radiative forcing derived in Smith et al. (2020) from a collection of CMIP6 models that performed idealized aerosol perturbation experiments. We also provided values from additional CMIP6 models that became available subsequent to its publication (as well as CMIP5 models previously reported, for completeness). The errors in the previous study resulted from two larger errors that, fortuitously, largely cancelled in the global mean, though for certain sub-components these errors do not cancel and are non-negligible. Most notably, the direct radiative forcing from absorbing aerosols averaged across CMIP6 models is more than 40% larger in the present study. Aerosol effective radiative forcings derived herein have negligible residuals and agree well with values derived using an independent double radiation call technique, both in the global mean and in geographic structure. Code to perform the accurate APRP method is provided at the link in the *Code availability* section.

*Code availability.* Code to perform the calculations in this study is available at https://doi.org/10.5281/zenodo.7809085 (Zelinka, 2021).

*Data availability.* All CMIP climate model data used in this study is available from the Earth System Grid Federation (https://esgf.llnl.gov/).

*Author contributions.* All analyses in the paper were performed by MDZ. The first draft of the manuscript was written by MDZ and all authors commented on subsequent versions of the manuscript. All authors read and approved the final manuscript.

*Competing interests.* The authors declare no competing interests.

*Acknowledgements.* We acknowledge the World Climate Research Programme, which, through its Working Group on Coupled Modelling,
coordinated and promoted CMIP. We thank the climate modeling groups for producing and making available their model output, the Earth
System Grid Federation (ESGF) for archiving the data and providing access, and the multiple funding agencies who support CMIP and ESGF.
MDZ, YQ, and KET were supported by the U.S. Department of Energy (DOE) Regional and Global Model Analysis program area. CJS was
supported by a NERC/IIASA Collaborative Research Fellowship (NE/T009381/1). The work of MDZ and KET and was performed under
the auspices of the U.S. DOE by Lawrence Livermore National Laboratory under Contract DE-AC52-07NA27344. The Pacific Northwest
National Laboratory is operated for the DOE by the Battelle Memorial Institute under Contract DE-AC05-76RL01830. We thank Susanne
Bauer for assistance interpreting the GISS results as well as two anonymous reviewers and editor Yuan Wang for helpful comments that
improved the manuscript.

## Appendix A

### A.1 Introduction

In this Appendix, we provide further explanation of the APRP technique and how it is to be used correctly. We then explain the two implementation errors in Smith et al. (2020), the first involving the calculation of albedo sensitivities and the second involving usage of incorrect SW fluxes at the top of atmosphere.

### A.2 APRP

As detailed in Taylor et al. (2007), the planetary albedo ($A$) can be written in terms of the surface albedo ($\alpha$), an atmospheric transmittance coefficient ($\mu$), and an atmospheric scattering coefficient ($\gamma$):

$$A = \mu\gamma + \frac{\mu\alpha(1-\gamma)^2}{1-\alpha\gamma} \tag{A1}$$

Surface albedo is computed as the ratio of upwelling to downwelling SW fluxes at the surface:

$$\alpha = SW^{\uparrow}_{SFC}/SW^{\downarrow}_{SFC}. \tag{A2}$$

Atmospheric transmittance is expressed as

$$\mu = A + \hat{Q}^{\downarrow}_{SFC}(1-\alpha), \tag{A3}$$

where

$$\hat{Q}^{\downarrow}_{SFC} = \frac{SW^{\downarrow}_{SFC}}{SW^{\downarrow}_{TOA}} \tag{A4}$$

is the ratio of surface to TOA incident SW flux, and the scattering coefficient is expressed as

$$\gamma = \frac{\mu - \hat{Q}^{\downarrow}_{SFC}}{\mu - \alpha\hat{Q}^{\downarrow}_{SFC}}. \tag{A5}$$

All of the terms given in A2 - A5 can be computed for both clear-sky fluxes and for overcast-sky fluxes. This separation into clear- and overcast-sky conditions, along with some assumptions, allow for the effects of clouds to be isolated from other atmospheric constituents (primarily aerosols in the present study). Specifically, the scattering coefficient in the overcast portion of a scene can be expressed as a combination of the scattering coefficients due to non-cloud constituents and the cloud itself:

$$(1-\gamma_{oc}) = (1-\gamma_{clr})(1-\gamma_{cld}), \tag{A6}$$

where subscripts $clr$ and $oc$ refer to clear- and overcast-sky conditions. Similarly, the transmissivity coefficients are related as:

$$\mu_{oc} = \mu_{clr}\mu_{cld}. \tag{A7}$$

This allows total planetary albedo to be expressed as the sum of clear-sky albedo scaled by the clear-sky portion of the scene and overcast-sky albedo scaled by the cloudy portion of the scene:

$$A = (1 - c)A_{clr} + cA_{oc}, \tag{A8}$$

where $c$ is the total cloud fraction. Hence, the planetary albedo is a function of seven parameters:

$$A = A(c, \alpha_{clr}, \alpha_{oc}, \mu_{clr}, \mu_{cld}, \gamma_{clr}, \gamma_{cld}). \tag{A9}$$

The derivation of these expressions (and the assumptions made in doing so) are detailed in Taylor et al. (2007), and so are not further explained here.

Having determined all of the terms upon which planetary albedo depends, we can now substitute in values of individual coefficients from the perturbed climate experiment to isolate their individual impacts on albedo. For example, to determine the impact on planetary albedo from the change in aerosol scattering, we difference the albedo computed with (only) $\gamma_{clr}$ taken from the perturbed experiment (indicated with superscript 'pert') and the albedo computed with all fields set to their control state (no superscript):

$$\frac{\partial A}{\partial \gamma_{clr}} \Delta \gamma_{clr} = A(c, \alpha_{clr}, \alpha_{oc}, \mu_{clr}, \mu_{cld}, \gamma_{clr}^{pert}, \gamma_{cld}) - A(c, \alpha_{clr}, \alpha_{oc}, \mu_{clr}, \mu_{cld}, \gamma_{clr}, \gamma_{cld}). \tag{A10}$$

In practice, we do this calculation twice – once as a forward calculation as shown above and once as a backward calculation in which all fields are set to their perturbed value and the isolated field is set to its control state – and then these are averaged.

### A.3 The Albedo Sensitivity Error in Smith et al. (2020)

Smith et al. (2020) did not use A9 to estimate their components but rather followed a different procedure. In the example considered above, they compute the impact on planetary albedo from the change in aerosol scattering as the sum of two terms:

$$\frac{\partial A}{\partial \gamma_{clr}} \Delta \gamma_{clr} = (1 - c)\frac{\partial A_{clr}}{\partial \gamma_{clr}} \Delta \gamma_{clr} + c\frac{\partial A_{oc}}{\partial \gamma_{clr}} \Delta \gamma_{clr}, \tag{A11}$$

where the responses of $A_{clr}$ and $A_{oc}$ to perturbations in $\gamma_{clr}$ are computed by substituting directly into Eq A1. The error arises for the latter term, which Smith et al. compute as:

$$\frac{\partial A_{oc}}{\partial \gamma_{clr}} \Delta \gamma_{clr} = A(\alpha_{oc}, \mu_{oc}, \gamma_{clr}^{pert}) - A(\alpha_{oc}, \mu_{oc}, \gamma_{clr}). \tag{A12}$$

Note that in A12 the *clear-sky* scattering coefficients are erroneously used to compute *overcast-sky* albedo. The correct calculation is:

$$\frac{\partial A_{oc}}{\partial \gamma_{clr}} \Delta \gamma_{clr} = A(\alpha_{oc}, \mu_{oc}, \gamma_{oc*}^{pert}) - A(\alpha_{oc}, \mu_{oc}, \gamma_{oc}), \tag{A13}$$

where we have defined $\gamma_{oc*}^{pert}$ via A6:

$$(1 - \gamma_{oc*}^{pert}) = (1 - \gamma_{clr}^{pert})(1 - \gamma_{cld}). \tag{A14}$$

This modified overcast-sky scattering coefficient allows changes in the clear-sky scattering coefficient to independently affect the overcast-sky scattering coefficient holding the cloud scattering fixed. Hence it incorporates the impact on atmospheric scattering under overcast-sky condtions from (solely) changes in clear-sky (aerosol) scattering, which is the coefficient needed to correctly estimate the influence of changes in clear-sky (aerosol) scattering on overcast-sky albedo. This is already implicitly done in the correct method involving substitution into Eq A10. It is only when substituting coefficients directly into Eq A1 (as done by Smith et al.) that one has to separately account for this.

Why does this error cause an overestimate of the sensitivity of TOA albedo to aerosols relative to the correct implementation of APRP? The difference between scattering coefficients in the erroneous expression (A12) is:

$$\Delta\gamma_{smith} = (\gamma_{clr}^{pert} - \gamma_{clr}). \tag{A15}$$

In contrast, after some algebra one can show that the difference between scattering coefficients in the corrected expression (A13) is:

$$\Delta\gamma_{true} = \gamma_{oc*}^{pert} - \gamma_{oc} = (1 - \gamma_{cld})(\gamma_{clr}^{pert} - \gamma_{clr}), \tag{A16}$$

which is equal to $\Delta\gamma_{smith}$ scaled by $1 - \gamma_{cld}$. This scaling factor, which is less than 1, represents the attenuation of aerosol effects by the presence of clouds: The larger the cloud scattering coefficient, the greater the attentuation and hence the weaker the aerosol influence on TOA albedo. Therefore the correct formulation of APRP, which accurately accounts for this attenuation, has a weaker sensitivity of TOA albedo to changes in clear-sky (aerosol) scattering. In other words, Smith's scattering coefficient changes are overestimated by the factor

$$\frac{\Delta\gamma_{smith}}{\Delta\gamma_{true}} = \frac{1}{1 - \gamma_{cld}} \tag{A17}$$

We note that similar coding errors are present in Smith et al.'s calculation of the sensitivity of overcast-sky albedo to aerosol absorption, cloud scattering, and cloud absorption. We find that the errors in the sensitivity of overcast-sky albedo to aerosol and cloud absorption are small in practice because cloud absorption of SW radiation is small. We do not detail these here for the sake of brevity.

### A.4 The Insolation Error in Smith et al. (2020)

All of the aforementioned APRP calculations yield sensitivities of planetary albedo ($A$) to perturbations in the seven parameters noted in A9. These need to be multiplied by the insolation at the TOA in order to determine the impact on the TOA SW energy budget. The net absorbed SW radiation at the TOA is expressed as

$$SW_{abs} = S(1 - A), \tag{A18}$$

where $S$ is the SW flux at the top of the atmosphere and $A$ is the planetary albedo. Its sensitivity to A is therefore

$$\frac{\partial SW_{abs}}{\partial A} = -S. \tag{A19}$$

In Smith et al.'s formulation, $S$ is

$$S_{smith} = (SW^{\downarrow}_{TOA} - SW^{\uparrow}_{TOA}) \tag{A20}$$

whereas it should be

$$S_{true} = SW^{\downarrow}_{TOA} \tag{A21}$$

By using the net (downwelling minus upwelling) SW flux, Smith et al's values of the TOA radiative impact are therefore underestimated by the factor

$$S_{smith}/S_{true} = 1 - \frac{SW^{\uparrow}_{TOA}}{SW^{\downarrow}_{TOA}} = 1 - A, \tag{A22}$$

which is roughly 70%.

## Appendix B

### B.1 Introduction

In this Appendix, we describe in greater detail how the various aerosol effective radiative forcings relate to each other and to the IPCC AR6 definitions (Forster et al., 2021). Specifically, we relate the LW ERFs defined by IPCC to the ERF proxies derived using standard model output in Section B.2. We then relate the IPCC SW and LW ERF proxies to the ERFs derived using the Ghan (2013) double-call method (Section B.3). Finally, the ERFs derived by the Ghan (2013) double-call method are related to those derived using the Taylor et al. (2007) APRP method (Section B.4) and to the LW ERF proxies derived using standard model output (Section B.5).

### B.2 LW ERFs: Proxies Derived from Standard Model Output

One can expand our expression for $ERF_{ari}^{P,LW}$ as:

$$ERF_{ari}^{P,LW} = \Delta R_{cs}^{LW} - \Delta R_{cs}^{T_0} = IRF_{ari,cs}^{LW} + K_{cs}^{T}\Delta T + K_{cs}^{q,LW}\Delta q. \tag{B1}$$

Given the IPCC definition of direct effective radiative forcing (Eq 2), we can express our proxy as

$$ERF_{ari}^{P,LW} = ERF_{ari}^{LW} - K^{C,LW}\Delta C_{semidirect} - M_{cld}^{LW}, \tag{B2}$$

where

$$M_{cld}^{LW} = (IRF_{ari}^{LW} - IRF_{ari,cs}^{LW}) + (K^{T} - K_{cs}^{T})\Delta T + (K^{q,LW} - K_{cs}^{q,LW})\Delta q. \tag{B3}$$

Therefore in the LW, our proxy for the direct effect equals IPCC's direct effect, minus the semidirect effect, minus masking terms that quantify how much the radiative impact of rapid changes in temperature, humidity, and aerosols are attenuated by the presence of clouds.

Turning now to the LW indirect effect, we note that the change in all-sky TOA net LW radiation is given by:

$$\Delta R^{LW} = IRF_{ari}^{LW} + K^{T}\Delta T + K^{q,LW}\Delta q + K^{C,LW}\Delta C. \tag{B4}$$

Therefore,

$$ERF_{aci}^{P,LW} = K^{C,LW}\Delta C + M_{cld}^{LW}, \tag{B5}$$

where $M_{cld}^{LW}$ is defined above. Putting this in terms of IPCC nomenclature:

$$ERF_{aci}^{P,LW} = ERF_{aci}^{LW} + K^{C,LW}\Delta C_{semidirect} + M_{cld}^{LW}. \tag{B6}$$

Therefore in the LW, our proxy for the indirect effect equals IPCC's indirect effect, plus the semidirect effect, plus masking terms that quantify how much the radiative impact of changes in temperature, humidity, and aerosols are attenuated by the presence of clouds. The sum of the direct and indirect LW effects are the same, independent of how the individual components are defined. Thus, from Eqs B2 and B6

$$ERF_{ari}^{P,LW} + ERF_{aci}^{P,LW} = ERF_{ari}^{LW} + ERF_{aci}^{LW}. \tag{B7}$$

### B.3 Double Radiation Call Method

#### B.3.1 Ghan's Direct Effect

Expanding Eq 14, one can express the change in TOA net radiation as

$$\Delta R = ERF_{ari}^G + K_{af}^T \Delta T + K_{af}^q \Delta q + K_{af}^\alpha \Delta \alpha + K_{af}^C \Delta C, \tag{B8}$$

or, equivalently,

$$\Delta R = IRF_{ari} + K^T \Delta T + K^q \Delta q + K^\alpha \Delta \alpha + K^C \Delta C \tag{B9}$$

Combining the previous two equations yields an expression for Ghan's $ERF_{ari}^G$ in terms of the true instantaneous aerosol direct forcing:

$$ERF_{ari}^G = IRF_{ari} - M_{aer}, \tag{B10}$$

where the aerosol masking is given by:

$$M_{aer} = (K_{af}^T - K^T)\Delta T + (K_{af}^q - K^q)\Delta q + (K_{af}^\alpha - K^\alpha)\Delta \alpha + (K_{af}^C - K^C)\Delta C. \tag{B11}$$

Therefore Ghan's direct aerosol radiative forcing equals IPCC's instantaneous direct forcing minus masking terms that quantify how much the radiative impact of changes in temperature, humidity, surface albedo, and clouds are attenuated by the presence of aerosols.

#### B.3.2 Ghan's Indirect Effect

Turning now to the indirect effect, let us write the change in TOA energy budget change as:

$$\Delta R = ERF_{ari}^G + ERF_{aci}^G + K_{af,cs}^T \Delta T + K_{af,cs}^q \Delta q + K_{af,cs}^\alpha \Delta \alpha. \tag{B12}$$

Defining

$$M_{aer,cld} = (K_{af,cs}^T - K^T)\Delta T + (K_{af,cs}^q - K^q)\Delta q + (K_{af,cs}^\alpha - K^\alpha)\Delta \alpha, \tag{B13}$$

which measures how much the radiative impact of changes in temperature, humidity, and surface albedo are masked by clouds and aerosols, and combining Eqs B9 and B11 yields an expression for the total cloud-induced radiation anomalies in terms of $ERF_{aci}^G$:

$$K^C \Delta C = ERF_{ari}^G + ERF_{aci}^G - IRF_{ari} + M_{aer,cld}. \tag{B14}$$

Given Eq B10, we can therefore write:

$$K^C \Delta C = ERF_{aci}^G - M_{aer} + M_{aer,cld} \tag{B15}$$

or equivalently,

$$K^C \Delta C = ERF_{aci}^G + (K_{af,cs}^T - K_{af}^T)\Delta T + (K_{af,cs}^q - K_{af}^q)\Delta q + (K_{af,cs}^\alpha - K_{af}^\alpha)\Delta\alpha + (K^C - K_{af}^C)\Delta C. \tag{B16}$$

We can now express Ghan's $ERF_{aci}^G$ in terms of IPCC's aerosol indirect forcing:

$$ERF_{aci}^G = ERF_{aci} + M_{aer} - M_{aer,cld} + K^C \Delta C_{semidirect}. \tag{B17}$$

Therefore Ghan's indirect effect equals IPCC's indirect effect plus masking terms that quantify how much the radiative impact of changes in temperature, humidity, and surface albedo are attenuated by the presence of clouds under aerosol-free conditions and how much the radiative impact of changes in clouds are attenuated by the presence of aerosols, plus the semidirect effect.

### B.3.3 Ghan's Other Forcing Term

Finally, let us separate the third forcing term defined by Ghan (2013) into its LW and SW components:

$$ERF_{other}^{G,LW} = K_{af,cs}^T \Delta T + K_{af,cs}^{q,LW} \Delta q, \tag{B18}$$

and

$$ERF_{other}^{G,SW} = K_{af,cs}^{q,SW} \Delta q + K_{af,cs}^\alpha \Delta\alpha. \tag{B19}$$

We can expand $ERF_{other}^{G,SW}$ as:

$$ERF_{other}^{G,SW} = \Delta R_{alb}^{SW} + K_{af,cs}^{q,SW} \Delta q + (K_{af,cs}^\alpha - K^\alpha)\Delta\alpha. \tag{B20}$$

Therefore $ERF_{other}^{G,SW}$ – which Ghan (2013) refers to as the surface albedo forcing – equals $\Delta R_{alb}^{SW}$ plus the aerosol-free clear-sky radiative contributions from changes in humidity, plus a masking term that quantifies how much the radiative impact of changes in surface albedo are attenuated by the presence of both clouds and aerosols.

## B.4 Relating SW ERF Terms: Ghan (2013) vs APRP

Combining Eqs 7 and B10 yields the relationship between Ghan- and APRP-derived SW direct radiative forcing:

$$ERF_{ari}^{A,SW} = ERF_{ari}^{G,SW} + M_{aer}^{SW} + K^{q,SW} \Delta q, \tag{B21}$$

Because $K^{C,SW}\Delta C$ is equivalent to $ERF_{aci}^{A,SW}$ (cf Eqs 5 and 7), a simple substitution into Eq B15 (applied to the shortwave) yields the relationship between the APRP and Ghan (2013) representations of the SW indirect effect:

$$ERF_{aci}^{A,SW} = ERF_{aci}^{G,SW} - M_{aer}^{SW} + M_{aer,cld}^{SW}. \tag{B22}$$

Similarly the relation between APRP's surface albedo component and Ghan's equivalent ($ERF_{other}^{G,SW}$) is already expressed in Eq B20:

$$\Delta R_{alb}^{A,SW} = ERF_{other}^{G,SW} - K_{af,cs}^{q,SW} \Delta q - (K_{af,cs}^\alpha - K^\alpha)\Delta\alpha. \tag{B23}$$

## B.5 Relating LW ERF Terms: Ghan (2013) vs Proxies

Subtracting Eqs 14 and 16 from Eq 11 and rearranging terms, we can write:

$$ERF_{ari}^{P,LW} = ERF_{ari}^{G,LW} + ERF_{other}^{G,LW} - \Delta R_{cs}^{T_0} + \epsilon, \tag{B24}$$

where

$$\epsilon = (\Delta R_{cs}^{LW} - \Delta R^{LW}) - (\Delta R_{af,cs}^{LW} - \Delta R_{af}^{LW}) = \Delta CRE^{LW} - \Delta CRE_{af}^{LW} \tag{B25}$$

is very small because the change in LW cloud radiative effect is roughly the same under aerosol-free and aerosol-present conditions. This means that our proxy for longwave direct effect equals the sum of Ghan's direct effect and "other" term plus adjustments that account for the radiative impact of surface temperature change and differences in the CRE response under aerosol-free and aerosol-present conditions. Combining Eqs B6 and B17 yields the relationship between the Ghan- and proxy-derived estimates of the LW indirect effect:

$$ERF_{aci}^{P,LW} = ERF_{aci}^{G,LW} - M_{aer}^{LW} + M_{aer,cld}^{LW} + M_{cld}^{LW}. \tag{B26}$$

In the LW, $M_{aer}$ is zero, so

$$ERF_{aci}^{P,LW} = ERF_{aci}^{G,LW} + M_{aer,cld}^{LW} + M_{cld}^{LW}. \tag{B27}$$

This means that our proxy for longwave indirect effect equals Ghan's indirect effect plus masking terms quantifying how strongly clouds attenuate the LW impact of changes in temperature, humidity, and aerosols and how strongly clouds and aerosols together attenuate the LW impact of changes in temperature and humidity.

**Table A1.** Climate model output used in this study. APRP requires the first eight fields. LW ERF components are estimated using the following two fields. These ten fields are routinely diagnosed in CMIP5 and CMIP6 models. For the double-call method, we rely upon aerosol-free radiative fluxes, which are the final four fields. These are only available for a subset of CMIP6 models. Aerosol-free upwelling SW radiation fluxes at the top of the atmosphere from the NorESM2-LM and NorESM2-MM models were found to be actually the *net* (downwelling minus upwelling) aerosol-free SW fluxes, and were corrected prior to usage.

| Description | Variable name |
|---|---|
| Total cloud fraction | clt |
| TOA downwelling SW radiation | rsdt |
| TOA upwelling SW radiation | rsut |
| TOA upwelling SW radiation under clear-sky conditions | rsutcs |
| Surface downwelling SW radiation | rsds |
| Surface downwelling SW radiation under clear-sky conditions | rsdscs |
| Surface upwelling SW radiation | rsus |
| Surface upwelling SW radiation under clear-sky conditions | rsuscs |
| TOA outgoing longwave radiation | rlut |
| TOA outgoing longwave radiation under clear-sky conditions | rlutcs |
| TOA upwelling SW radiation under aerosol-free conditions | rsutaf |
| TOA upwelling SW radiation under aerosol-free clear-sky conditions | rsutcsaf |
| TOA outgoing LW radiation under aerosol-free conditions | rlutaf |
| TOA outgoing LW radiation under aerosol-free clear-sky conditions | rlutcsaf |

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

**Table A2.** Abbreviations commonly used in this study.

| Abbreviation | Description |
|---|---|
| $\Delta$ | Change between pre-industrial and present-day |
| $\alpha$ | Surface albedo |
| APRP | Approximate partial radiative perturbation |
| $C$ | Cloud fraction |
| $CRE$ | Cloud radiative effect |
| $ERF$ | Effective radiative forcing |
| $ERF_{aci}$ | $ERF$ due to aerosol-cloud interactions |
| $ERF_{aci,abs}$ | $ERF_{aci}$ due to cloud absorption |
| $ERF_{aci,amt}$ | $ERF_{aci}$ due to cloud amount |
| $ERF_{aci,scat}$ | $ERF_{aci}$ due to cloud scattering |
| $ERF_{ari}$ | $ERF$ due to aerosol-radiation interactions |
| $ERF_{ari,abs}$ | $ERF_{ari}$ due to aerosol absorption |
| $ERF_{ari,scat}$ | $ERF_{ari}$ due to aerosol scattering |
| $IRF$ | Instantaneous radiative forcing |
| $K^{\chi}$ | Sensitivity of top of atmosphere radiation to $\chi$ |
| $LW$ | Longwave radiation |
| $q$ | Specific humidity |
| $R$ | Net (downwelling minus upwelling) top of atmosphere radiation |
| $\Delta R_{alb}$ | Change in top of atmosphere radiation due to surface albedo changes |
| $SW$ | Shortwave radiation |
| $T$ | Temperature |
| $TOA$ | Top of atmosphere |
| $A$ | Superscript indicating value was computed via APRP technique |
| $G$ | Superscript indicating value was computed via Ghan (2013) technique |
| $P$ | Superscript indicating value is a proxy for ERF derived from standard model output |
| $af$ | Subscript indicating aerosol-free radiative fluxes |
| $cs$ | Subscript indicating clear-sky radiative fluxes |

Ghan, S. J.: Technical Note: Estimating aerosol effects on cloud radiative forcing, Atmospheric Chemistry and Physics, 13, 9971–9974, https://doi.org/10.5194/acp-13-9971-2013, 2013.

Gryspeerdt, E., Mülmenstädt, J., Gettelman, A., Malavelle, F. F., Morrison, H., Neubauer, D., Partridge, D. G., Stier, P., Takemura, T., Wang, H., Wang, M., and Zhang, K.: Surprising similarities in model and observational aerosol radiative forcing estimates, Atmospheric Chemistry and Physics, 20, 613–623, https://doi.org/10.5194/acp-20-613-2020, publisher: Copernicus GmbH, 2020.

Huang, Y., Xia, Y., and Tan, X. X.: On the pattern of CO2 radiative forcing and poleward energy transport, Journal of Geophysical Research-Atmospheres, 122, 10 578–10 593, https://doi.org/10.1002/2017jd027221, 2017.

Miller, R. L., Schmidt, G. A., Nazarenko, L. S., Bauer, S. E., Kelley, M., Ruedy, R., Russell, G. L., Ackerman, A. S., Aleinov, I., Bauer, M., Bleck, R., Canuto, V., Cesana, G., Cheng, Y., Clune, T. L., Cook, B. I., Cruz, C. A., Del Genio, A. D., Elsaesser, G. S., Faluvegi, G., Kiang, N. Y., Kim, D., Lacis, A. A., Leboissetier, A., LeGrande, A. N., Lo, K. K., Marshall, J., Matthews, E. E., McDermid, S., Mezuman, K., Murray, L. T., Oinas, V., Orbe, C., Pérez García-Pando, C., Perlwitz, J. P., Puma, M. J., Rind, D., Romanou, A., Shindell, D. T., Sun, S., Tausnev, N., Tsigaridis, K., Tselioudis, G., Weng, E., Wu, J., and Yao, M.-S.: CMIP6 Historical Simulations (1850–2014) With GISS-E2.1, Journal of Advances in Modeling Earth Systems, 13, e2019MS002 034, https://doi.org/10.1029/2019MS002034, _eprint: https://onlinelibrary.wiley.com/doi/pdf/10.1029/2019MS002034, 2021.

Persad, G. G., Samset, B. H., and Wilcox, L. J.: Aerosols must be included in climate risk assessments, Nature, 611, 662–664, https://doi.org/10.1038/d41586-022-03763-9, bandiera_abtest: a Cg_type: Comment Number: 7937 Publisher: Nature Publishing Group Subject_term: Climate change, Atmospheric science, Policy, 2022.

Pincus, R., Forster, P. M., and Stevens, B.: The Radiative Forcing Model Intercomparison Project (RFMIP): experimental protocol for CMIP6, Geoscientific Model Development, 9, 3447–3460, https://doi.org/10.5194/gmd-9-3447-2016, 2016.

Rotstayn, L. D., Collier, M. A., Chrastansky, A., Jeffrey, S. J., and Luo, J. J.: Projected effects of declining aerosols in RCP4.5: unmasking global warming?, Atmos. Chem. Phys., 13, 10 883–10 905, https://doi.org/10.5194/acp-13-10883-2013, 2013.

Sherwood, S. C., Webb, M. J., Annan, J. D., Armour, K. C., Forster, P. M., Hargreaves, J. C., Hegerl, G., Klein, S. A., Marvel, K. D., Rohling, E. J., Watanabe, M., Andrews, T., Braconnot, P., Bretherton, C. S., Foster, G. L., Hausfather, Z., Heydt, A. S. v. d., Knutti, R., Mauritsen, T., Norris, J. R., Proistosescu, C., Rugenstein, M., Schmidt, G. A., Tokarska, K. B., and Zelinka, M. D.: An Assessment of Earth's Climate Sensitivity Using Multiple Lines of Evidence, Reviews of Geophysics, 58, e2019RG000 678, https://doi.org/https://doi.org/10.1029/2019RG000678, 2020.

Smith, C. J., Kramer, R. J., Myhre, G., Alterskjær, K., Collins, W., Sima, A., Boucher, O., Dufresne, J.-L., Nabat, P., Michou, M., Yukimoto, S., Cole, J., Paynter, D., Shiogama, H., O'Connor, F. M., Robertson, E., Wiltshire, A., Andrews, T., Hannay, C., Miller, R., Nazarenko, L., Kirkevåg, A., Olivié, D., Fiedler, S., Lewinschal, A., Mackallah, C., Dix, M., Pincus, R., and Forster, P. M.: Effective radiative forcing and adjustments in CMIP6 models, Atmospheric Chemistry and Physics, 20, 9591–9618, https://doi.org/10.5194/acp-20-9591-2020, publisher: Copernicus GmbH, 2020.

Taylor, K. E., Crucifix, M., Braconnot, P., Hewitt, C. D., Doutriaux, C., Broccoli, A. J., Mitchell, J. F. B., and Webb, M. J.: Estimating Shortwave Radiative Forcing and Response in Climate Models, J. Climate, 20, 2530–2543, https://doi.org/10.1175/JCLI4143.1, 2007.

Watson-Parris, D. and Smith, C. J.: Large uncertainty in future warming due to aerosol forcing, Nature Climate Change, 12, 1111–1113, https://doi.org/10.1038/s41558-022-01516-0, number: 12 Publisher: Nature Publishing Group, 2022.

Wilcox, L. J., Highwood, E. J., and Dunstone, N. J.: The influence of anthropogenic aerosol on multi-decadal variations of historical global climate, Environmental Research Letters, 8, 024 033, https://doi.org/10.1088/1748-9326/8/2/024033, 2013.

Zelinka, M.: mzelinka/aprp: Sep 17, 2021 Release, https://doi.org/10.5281/zenodo.5514142, 2021.

Zelinka, M. D., Andrews, T., Forster, P. M., and Taylor, K. E.: Quantifying components of aerosol-cloud-radiation interactions in climate models, Journal of Geophysical Research-Atmospheres, 119, 7599–7615, https://doi.org/10.1002/2014jd021710, 2014.