# Peer review of "Comparison of Methods to Estimate Aerosol Effective Radiative Forcings in CMIP Models"

_EGUsphere, 2023_

## Author Response (AR1)

**Response to Reviewers**

**Aerosol Effective Radiative Forcings in CMIP Models**
**by Zelinka et al.**
**egusphere-2023-689**

**Response to Anonymous Referee #1:**

In their manuscript, the authors revisit existing methods to decompose the aerosol forcing into different terms and compare them against each other. In particular they flag some errors in the implementation of the APRP method by Smith et al. (ACP, doi: 10.5194/acp-20-9591-2020, 2020). As Chris Smith is a co-author of this manuscript, I assume he agrees with this! Overall this is a technical yet interesting manuscript that also documents the aerosol forcings of CMIP5 and CMIP6 models. I recommend publication after my comments below are accounted for.

We are glad to hear that the reviewer found the manuscript interesting.  We thank the reviewer for carefully reading the manuscript and providing these helpful comments, which are each addressed in turn below.

**Comments (major and minor interspersed)**

Title: I did not find the title to be very informative. May I suggest instead "On methods to estimate aerosol effective forcings in CMIP models" or something that conveys the idea that several methods are looked at and compared in this manuscript?

We have changed the title to "Comparison of Methods to Estimate Aerosol Effective Radiative Forcings in CMIP Models."

Abstract and possibly elsewhere, "forcing from aerosol scattering": By scattering solar radiation, aerosols generally tend to increase the mean photon path and therefore gaseous absorption. I would argue that "forcing due to aerosol scattering" or "forcing induced by aerosol scattering" is a more appropriate term than "forcing from aerosol scattering". Likewise for cloud scattering and absorption.

"From" has been changed to "due to."

Equation 1: there is an additional term in this equation as ΔR also depends on the climate response over land (SST are fixed in these experiments but Land Surface Temperatures or LST are not). This is well explained in e.g. Sherwood et al (BAMS, 2015) as, I'm sure, the authors are aware. I would suggest that this additional term is included in Eq. 1. How it can be corrected or reasons why it can be neglected should be discussed.

Thank you for pointing this out.  We have reformulated our expressions to more closely adhere to the standard convention used in the community (including IPCC AR6).  Specifically, we have modified Eq 1 to make it clear that both the surface albedo response and the surface temperature responses are not part of the ERF.  We add some discussion

of this point as well. To be consistent with this, we also update our LW ERF_ari formulation to exclude LW radiative effects arising from land surface temperature changes. This involved additional analysis in which we estimated the impact on clear-sky TOA LW fluxes from surface temperature changes using surface radiative kernels. These are very small corrections, on the order of 0.05 W/m2 globally, consistent with Table 5 of Smith et al (2020). This also impacts our comparison to the double-call LW ERF values shown in Figure 8 and detailed mathematically in Appendix B. The upshot of this is that our LW ERF_ari values are smaller (less positive) in the revised manuscript, making the overall net negative ERF about 5% stronger, consistent with the 5% inflation that IPCC AR6 applied to the ERF values derived in Smith et al (2020) and Zelinka et al (2014).

Line 78: is "temperature" here meant to be "atmospheric temperature" or both "atmospheric" and "surface" temperature? See my comment above on Equation 1.
We now state explicitly what is meant by this, as part of the revisions done in response to the previous comment.

Line 78 and elsewhere: IPCC AR6 is too vague, please cite explicitly the relevant Chapter or Chapters.
The IPCC AR6 chapter (Forster et al 2021) was cited 6 lines prior, so we felt that it was not necessary to cite it again here. However, in several subsequent references to IPCC AR6, we include the citation again. This citation is now present 3 times in the paper.

Equation 5: Aerosols may also change surface albedo by changing the distribution between direct and diffuse radiation. Indeed albedo is not an intrinsic property of the surface but depends on the properties of the incoming solar radiation. Several models now include different albedoes for direct and diffuse radiation, in particular over the ocean. By increasing the fraction of diffuse radiation at the surface in clear sky and by changing the amount of cloudiness, aerosols may thus modify the surface albedo. This additional term is probably small, and maybe negligible, but worth mentioning.
Thank you for pointing this out. We now mention this contribution to the surface albedo change.

Line 115: should it be IPCC ari and aci rather than direct and indirect effects?
It is correctly written as the indirect and semidirect effects. The semidirect effect is a sub-component of the ERF$_{ari}$ so it would not be correct to replace "semidirect" with "ari." The sentence is describing in words the equations that immediately precede it; because of this we favor "indirect effect" rather than the more cryptic "ERF$_{aci}$".

APRP method, section 2.2.2: I would strongly encourage the authors to include a bit more description of the APRP method so that this manuscript can be read as a stand alone piece without the need to refer back to Taylor et al (2007) to make sense of it.
We now include an entirely new Appendix A that details the APRP method and how Smith et al (2020) incorrectly implemented it. Because of this addition, and our desire to not

overwhelm more casual readers with APRP details in the main text, we have kept the description of APRP in Section 2.2.2 at the conceptual level and now simply refer any reader interested in further details to Appendix A.

Equation 15: although there is no change in notation, it may be worth repeating here that ΔR is the difference in R between two simulations.
We now remind the reader of this.

Line 160: see my comments above. Does it mean that the ΔR term due to the change in LST is negligible? Can you provide evidence of this?
The change in LST may not be negligible, but as this will not directly impact SW radiation, it will not degrade the ability of APRP to match the model-produced changes in radiation.

Lines 165-169: this paragraph needs more context to be understood. A more detailed section 2.2.2 would help the reader to appreciate fully this paragraph.
These two errors are now fully described in Appendix A, to which we refer the reader.

Figure 1: It may be worth saying in the caption that each circle represents a different CMIP model.
Added as suggested.

Section 3.3: It may be worth saying that the forcing year is different for CMIP5 and CMIP6 (2014 for the latter, I'm not sure for the former). However this should not have much of an impact as the aerosol forcing is thought to be pretty flat over that period.
These years were noted in Section 2.1, but we agree that it is helpful to explicitly note them again here, which we now do. We have also added this information to the captions of Tables 1 and 2 where the CMIP5 and CMIP6 ERFs are provided.

A Table summarizing all the symbols / notations used in the manuscript would be useful.
We have added a new Table A2 in the appendix containing the symbols, abbreviations, and other notation that are commonly used throughout the paper.

**Response to Anonymous Referee #2:**

**Summary:**

In their manuscript, the authors reveal biases in previously published aerosol effective radiative forcing estimates in CMIP6 models (Smith et al., 2020; doi: 10.1002/2014JD021710) which arise from two coding errors in the application of the approximate partial radiative perturbation (APRP) technique proposed by Taylor (2007; doi: 10.1175/JCLI4143.1). The authors show the impact of each of these errors on aerosol ERF components, correct them and compare the results with the ones derived using the classic double radiation call method (Ghan, 2013; doi: 10.5194/acp-13-9971-2013). Further, they describe, compare and relate the various aerosol

radiative forcing terms from the two approaches and list the corrected estimates of aerosol ERF and its components for CMIP6 and CMIP5 models.

**General Comments:**

Since this manuscript revisits existing methods to compute aerosol ERF, I find it particularly useful that they disassemble the aerosol ERF defined in IPCC AR6 and show how the various components relate to each other. Further they provide a detailed comparison of the various forcing terms and components from the two methods mentioned above, explaining differences and similarities in definitions and nomenclatures which is a great help to understand the involved processes and to be able to make meaningful comparisons.

We are glad to hear that the reviewer had a favorable view of our manuscript and found our explanation of the components comprising the various ERF definitions particularly useful.

Overall, the manuscript is well structured and logically. They start with describing the two coding errors in APRP application and demonstrating their impact. I am interested on how these coding errors emerged, and if you did something to the code to adjust this? Did you e.g. make the code more user friendly so that this does not happen again when users get your code from Zenodo? Otherwise, in this section, my main concerns were just a few explanations and figures to be adjusted.

Smith et al (2020) coded up the APRP analysis independently, and in doing so made some errors. Their code was not based on the code used in Zelinka et al (2014) that we made available on github and zenodo, which is essentially the same as the code used in this study. That said, the present code has been modified slightly from what was available on github prior to writing this paper. We have made it more user friendly by restructuring the code in several places, providing better comments, rewriting several functions in xarray (a more modern and better supported python package), and providing a jupyter notebook demonstrating its use. This refactoring has simplified and improved the usability and readability of the code, but its overall functionality is identical to what existed prior to this paper.

Then they compare the results with the standard double call method and emphasize the good agreement and usefulness of APRP. This could be shortened since this was shown in previous studies. However, since APRP only works in the SW, they used proxies for the LW components to be compared. Even though this has nothing to do with correcting Smith et. al. (2020), it is nice that they included this to give their study a complete picture and to be able to compute the net aerosol ERF.

We believe that the comparison of APRP to the double call method is important to demonstrate here, and is of appropriate length, so we have kept that unchanged. We are glad to hear that the reviewer appreciates our inclusion of LW results.

Finally they conclude with listing the corrected APRP results for all CMIP6 models and even CMIP5 models. In my opinion this part is the main point of this manuscript. The corrected

estimates should be compared to the biased CMIP6 estimates, as well as with CMIP5. The latter has been too brief and should be given more weight.

Below we provide a comparison between CMIP6 multi-model mean SW TOA values estimated by Smith et al (2020) and estimated in this study. The numbers for Smith below differ from those in Table 6 of Smith et al (2020) because we include four additional models that are now available (ACCESS-ESM1-5, BCC-ESM1, IPSL-CM6A-LR-INCA, MPI-ESM-1-2-HAM) and we exclude EC-Earth3 for the reasons described in the text. This figure is now included in the paper (new Figure 5), and we have added a paragraph noting the key differences between the two estimates.

[Figure]

We also now provide an additional paragraph discussing differences in the multi-model mean ERF values between CMIP5 and CMIP6, with a caution that "present-day" refers to different periods between the two MIPs (2000 for CMIP5 and 2014 for CMIP6), and that the fraction of models that represent aerosol effects on cloud optical properties and/or lifetimes may differ. To support this comparison, we added a new panel to Figure 9 containing the CMIP5 mean ERF values. We have done this calculation in two ways: once using all available models from both collections, and once using only models from the same modeling center between CMIP5 and CMIP6. The pairings are: CanESM2 / CanESM5; GFDL-CM3 / GFDL-CM4; HadGEM2-A / HadGEM3-GC31-LL; IPSL-CM5A-LR / IPSL-CM6A-LR; MIROC5 / MIROC6; MPI-ESM-LR / MPI-ESM-1-2-HAM; MRI-CGCM3 / MRI-ESM2-0; NorESM1-M / NorESM2-MM; bcc-csm1-1 / BCC-ESM1. We find that the differences hold regardless of whether the full collection of models or subset of models is considered. In the figure below we show these two comparisons.

[Figure]

I can recommend to publish this manuscript after minor revisions.
We thank the reviewer for carefully reading the manuscript and providing these helpful comments. The remaining comments are each addressed in turn below.

**Specific Comments:**

Title: I suggest to add something like "Correction to" to the title to make clear that you update something that is already published
We have changed the title to "Comparison of Methods to Estimate Aerosol Effective Radiative Forcings in CMIP Models."

**Abstract:**

L.7 – 10: You need to be more specific. I suggest to add "CMIP6 models" and "two coding errors" to hint where the bias is coming from.
Changed as suggested.

L. 10: The word "ground-truth" is a bit too strong for my taste – we are still talking of a model here, not reality. I suggest to reword.
We changed this phrase to "benchmark"

I would delete the last sentence and instead add some main conclusion on corrected CMIP6 vs biased CMIP6 and/or CMIP5 estimates.
We respectfully disagree with deleting the last sentence, which reassures readers that the APRP technique is both accurate and efficient, and provides insights about aerosol forcing that one cannot get any other way. We also resist elevating the CMIP5-to-CMIP6 differences to the level of the abstract because this comparison is misleading given (1) the different definitions of "present-day" between the two ensembles and (2) the different fraction of models that represent indirect effects.  However, we now provide a bit more detail in the abstract about the corrected vs biased ERF results than what we had previously.

**Text:**

L.20: "cease"? Isn't that too strong? Why should they do that? Maybe change to "decrease"
This "zero emissions commitment" is a standard calculation in climate science and is the scenario explored in the cited study. It is not meant to be realistic but rather a measure of how much warming we have already committed to based on cumulative emissions to date and assuming negative emissions technologies are not implemented. Nevertheless, it remains true that committed warming – however defined – is strongly affected by aerosol forcing so we have changed "cease" to "decrease".

L. 40: "If a small degradation of absolute accuracy can be tolerated" Like how much?
This phrase no longer appears.

L. 45: "made different choices that have quantitative impacts on the results" What choices? Please be more elaborate on this point in the "Data and Methods" Section, since this is what has led to the erroneous results.

We now add "(as described below)" to this line so that readers know we will explain this later. We don't feel that it is appropriate to describe the choices at this point in the introduction where we are giving a basic overview of the paper.

L. 165 – 172: Can you relate the two features to an equation already given in this manuscript? E.g. is Kα in EQ. 5 & 10 the albedo sensitivity term that has not been scaled correctly in Smith et al. (2020)? Can you name a specific code parameter or module that needed correction?
These two errors are now fully described in Appendix A, to which we refer the reader.

L. 209: "whereas the ERFalb SW bias vanishes because it is not affected by the albedo sensitivity formulation error" Can you explain this to me? How is it related to Eq. 5 & 10? Can you also state where the grey markers are in Fig. 3B? I assume they are under the blue ones, because correcting the albedo sens. has no effect?
We now make it clearer that it is the sensitivity of planetary albedo to atmospheric scattering and absorption parameters that is incorrect in Smith et al (2020), whereas the sensitivity of planetary albedo to surface albedo is implemented correctly. Hence the only error relevant for ERF$_{alb}$ is the insolation error.

We now note explicitly in the caption of Figure 3 that "gray markers in panel d are overlain by cyan markers". (Panel b is now panel d in this figure.)

L. 285: You state that you did the same as in Zelinka et al. (2014), so I assume you recomputed all values? So a difference by one in the second decimal place is ok. How come the difference in net ari+aci for MRI-CGCM3 is 0.03 (-1.16 vs -1.13)? See and cite! Zelinka et al. (2014), Table 1
Correct, we recomputed all values in this study and they are essentially identical to those reported in Zelinka et al (2014).  It is hard to track down why there are some differences at the 2nd decimal place.  A possibility is that our code from circa 2014 handled area-weighted global means slightly differently or that the CMIP data itself has been updated since then.

**Figures:**

Fig. 2, 3 & 4: I know you want to focus on the bias if this or that is wrongly implemented. However I find it a bit confusing. So if you show Smith et al as being the wrong graph, then why don't you show what would happen if you correct this or that? So, instead of writing e.g. "w/erroneous insolation" I suggest to write "w/ corrected albedo sens." Instead of going backwards to the erroneous results, you can go forward from erroneous to correct result. This means of course, to logically turn around graphs and corresponding text. Please discuss if this could be a better and more logical presentation, it is not a must-do!
Thanks for this suggestion. We had considered presenting the results in this way initially but prefer the version in the submitted manuscript for the following reasons. First, the present version explicitly shows the compensating errors in Smith's implementation. For example in Figure 3a-c one can see that the insolation error leads to underestimated ERF values but that the albedo sensitivity error leads to overestimated ERF values; together they almost perfectly compensate such that Smith's values lie very close to the 1:1 line. In contrast, in Figure 3d one

can see that the insolation error is not compensated by the albedo sensitivity error, so Smith's $\Delta SW_{alb}$ term remains biased.  Secondly, we want to avoid giving the impression that we start our analysis from the flawed Smith implementation of APRP and fix it, allowing us to arrive at the corrected values presented in this paper. Finally, if one prefers, one can think of "w/erroneous insolation" as identical to "w/corrected albedo sensitivity" as noted in the caption of Figure 3.

Technical Comments:

L. 39: "Fortunately, aerosol direct and indirect effects primarily operate in the SW with much smaller effects in the longwave (LW)," Please give some references for this!
References have been added.

L. 49: change to "APRP-derived"
Fixed.

L. 79: "cloud particle number " I would reword to "cloud droplet number", otherwise it is not clear whether you include aerosols or not
Changed to "cloud liquid and ice particle number concentrations and sizes".

EQ. 8: Is this the same as EQ. 6, just for SW?
Yes. We believe that keeping both is needed for clarity. (These are now Eqs 7 and 5.)

Fig. 1: Are all the red points overlain? Maybe you could state that somewhere, so that readers don't get confused and don't think you might have just used one model. Caption: Delete "true", Figure text: change to "TOA SW Residuals"
Changes made as suggested.

L. 180: Maybe state as in-line equation instead of text
We prefer to keep this as written.

L. 215: change to "clouds scatter more SW radiation than they absorb".
Changes made as suggested.

L. 240: make clear you mean the corrected APRP from this study and Ghan's double call
We have rephrased this sentence to be "The multi-model mean maps show excellent agreement in the spatial structure of each component between the correct implementation of APRP and Ghan (2013) methodologies, albeit with quantitative differences (Figure 7)."

L.256: "opposite-signed errors" change to "opposite-signed differences", errors are between Smith (2020) – APRP and this study APRP, but these I wouldn't state as errors
Changes made as suggested.

Fig. 3(a,b,c), Fig. 4(a,c,d), Fig. 5B: Please make the scaling equal between x and y axis. If there is not enough space on the x axis, reduce ticking but start and end with the same numbers as on y-axis. This way is a bit confusing.

The scaling was already equal between the x- and y-axes, but we now enforce that the ticks are also identical between axes.

Fig. 5C: Make grey line dashed.
Change made as suggested. (This is now Figure 6.)

Fig. 6: Insert the same color bar between column 2 and 3 and adjust the rightmost color bar (e.g. ranging from -2 to 2) for the difference plot. Please state that for Fig. 6 only the corrected APRP from this study is used.
We have modified the figure so that the first two columns have a different colorbar than the third (difference) column. As noted above in the modification to L240, when we first refer to this figure, we now clarify that this is the correct implementation of APRP. (This is now Figure 7.)

Fig. 7: Extend dashed grey 1:1 line all the way through the plot just as in the other plots.
Change made as suggested. (This is now Figure 8.)

Fig. 8: instead of "Amount" write "Cloud amount", change title to "Corrected CMIP6 mean aerosol ..."
We have changed all of the $ERF_{aci}$ bar labels to include a "Cloud" prefix and all of the $ERF_{ari}$ bar labels to include an "Aerosol" prefix. We have opted not to change the title since we do not think anyone would assume we are presenting the uncorrected values here. (This is now Figure 9.)